# Human uniqueness? Life history diversity among small-scale societies and chimpanzees

**Raziel J. Davison** [1,2]*, **Michael D. Gurven**[1,2]

**1** Integrative Anthropological Sciences, Department of Anthropology, University of California, Santa Barbara, Santa Barbara, CA, United States of America, **2** Broom Center for Demography, University of California, Santa Barbara, Santa Barbara, CA, United States of America

* razieldavison@gmail.com

## Abstract

### Background

Humans life histories have been described as "slow", patterned by slow growth, delayed maturity, and long life span. While it is known that human life history diverged from that of a recent common chimpanzee-human ancestor some ~4–8 mya, it is unclear how selection pressures led to these distinct traits. To provide insight, we compare wild chimpanzees and human subsistence societies in order to identify the age-specific vital rates that best explain fitness variation, selection pressures and species divergence.

### Methods

We employ Life Table Response Experiments to quantify vital rate contributions to population growth rate differences. Although widespread in ecology, these methods have not been applied to human populations or to inform differences between humans and chimpanzees. We also estimate correlations between vital rate elasticities and life history traits to investigate differences in selection pressures and test several predictions based on life history theory.

### Results

Chimpanzees' earlier maturity and higher adult mortality drive species differences in population growth, whereas infant mortality and fertility variation explain differences between human populations. Human fitness is decoupled from longevity by postreproductive survival, while chimpanzees forfeit higher potential lifetime fertility due to adult mortality attrition. Infant survival is often lower among humans, but lost fitness is recouped via short birth spacing and high peak fertility, thereby reducing selection on infant survival. Lastly, longevity and delayed maturity reduce selection on child survival, but among humans, recruitment selection is unexpectedly highest in longer-lived populations, which are also faster-growing due to high fertility.

### Conclusion

Humans differ from chimpanzees more because of delayed maturity and lower adult mortality than from differences in juvenile mortality or fertility. In both species, high child mortality

**Data Availability Statement:** Data are included as .xlsx files in Supporting Infromation (S2, S3 Tables).

**Competing interests:** The authors have declared that no competing interests exist

reflects bet-hedging costs of quality/quantity tradeoffs borne by offspring, with high and variable child mortality likely regulating human population growth over evolutionary history. Positive correlations between survival and fertility among human subsistence populations leads to selection pressures in human subsistence societies that differ from those in modern populations undergoing demographic transition.

## Introduction

Humans and chimpanzees, whose recent common ancestor dates to 4–8 million years ago [1, 2], share behavioral adaptations and life history traits that distinguish them from other primates [3, 4]. Human fertility schedules are similar to chimpanzees except for menopause, which appears unique among mammals, apart from a few toothed whale species [5]. Human fertility declines well in advance of survival, whereas reproductive and actuarial senescence appear to occur together in chimpanzees [6]. In addition, mortality profiles of modern hunter-gatherers are closer to chimpanzees than they are to today's low-mortality post-industrialized populations [7], but there is much variation among human and chimpanzee life histories [8, 9]. Despite this variation, primates are generally viewed as falling along the slow end of a slow-fast life history continuum [10] due to delayed maturity, longevity and relatively low fertility [11], with humans at the slowest end of primates [12]. Here, we evaluate human uniqueness by identifying the vital rates that drive life history variation among populations within each species as well as between species. In doing so, we also characterize the tradeoffs [13] that may have shaped human life history evolution.

Whereas many contemporary small-scale societies are growing rapidly [14], documented chimpanzee populations are typically shrinking, though favorable conditions promote increase in some wild groups [15, 16]. With the largest high-quality dataset assembled to date on fertility and mortality among human subsistence societies and chimpanzees [17], we employ life table response experiments [LTREs, 18, 19] to quantify the importance of particular age classes for driving population growth and to identify the vital rates that best explain the divergence of human and chimpanzee life histories. As a mathematical tool designed to decompose population growth rates into age-specific demographic components, LTREs are typically used to aid conservation efforts for endangered species. To our knowledge, this is the first application of LTREs to humans.

Although demographic patterns are well-described for many primate species, including chimpanzees [20–22], we provide a comprehensive, up-to-date and timely comparison of human and chimpanzee life histories using several metrics designed to assess the fitness importance of survival and fertility at different ages: *fitness contributions* illustrate how life history event schedules drive observed population fitness differences within and between species, while *fitness elasticities* reflect the force of selection and highlight the potential for fitness contributions if vital rates vary across populations [23]. Elasticities provide *prospective* predictions of the potential for vital rate effects on fitness (% change in fitness expected due a % change in a vital rate), but they do not always predict the most important vital rates that actually explain population-level differences in fitness [24]. Observed population-level fitness differences are estimated *retrospectively* through LTREs, which decompose population-level differences on the basis of observed differences in vital rates [24, 25].

Previous human-chimpanzee comparisons have used fewer populations, focused primarily on composite fertility and longevity differences, and have not systematically quantified species

life history differences [8, 22]. Because species comparisons are complicated by within-species life history variation [21], we: (a) compare species mean life histories (hereafter referred to as "composite" life histories), and (b) characterize the population drivers among populations of the same species. Using the average hunter-gatherer life history (HG) as a common reference, we compare hunter-gatherers with other natural fertility subsistence societies and with seven chimpanzee populations, including captive and managed populations representing chimpanzee "best case" life histories. We also compare the hunter-gatherer reference to three other composite life histories representing the average life histories calculated across non-exclusive foragers, and across chimpanzee populations exhibiting decreasing vs. increasing population growth. Without relying on model life tables or indirect demographic methods for age assignment or vital rate estimation, our dataset includes ten small-scale societies (five hunter-gatherers, three forager-horticulturalists, a pastoralist society and an "acculturated" hunter-gatherer population with some plant domestication) and seven chimpanzee populations with high quality fertility and mortality data (five wild, one managed and one captive).

We identify the vital rates that are most important in driving population growth and decline using fixed-effect LTREs [18, 19], which decompose contributions of different vital rates to observed differences in population growth rates. Vital rate contributions ($C_{ij}$) are estimated by multiplying vital rate sensitivities ($s_{ij}$), which reflects the fitness effect of a one-unit change in matrix element $a_{ij}$, by population-level differences ($\Delta a_{ii}$) in vital rates ($\Delta a_{ij} = a_{ij}^{(m)} - a_{ij}^{(R)}$; $C_{ij} = s_{ij} \Delta a_{ii}$), comparing each observed population ($m$) with a common reference ($R$). In our analysis, $R$ refers to the reference HG life history characterized by mean vital rates calculated across the five hunter-gatherer populations. We compare these results with the more familiar and widely-used elasticity analyses that prospectively estimate the potential for fitness effects of vital rates [19, 23]. Differences between realized fitness contributions from LTREs and the potential suggested by elasticities may indicate constraints on life history evolution. For instance, if stabilizing selection reduces variation in important vital rates [26], fitness contributions of high-elasticity rates are likely to be small [27]. More generally, when elasticities overestimate fitness effects this may reflect constraints on the stabilizing selection that would otherwise reduce variation in these important vital rates, whereas underestimation implies that vital rate differences are more important than *a priori* predictions from the force of selection. We also evaluate three predictions of life history theory based on fitness elasticities: (**P1**) survival, and especially juvenile recruitment (early infant survival $p_0$), should have the largest effect on population growth [28]; (**P2**) because both fertility and survival have positive elasticities, intrinsic population growth rates ($r$) should be greater in populations with higher life expectancy ($e_0$) and with higher total fertility rate (*TFR*); (**P3**) elasticity to child survival should be negatively correlated with life expectancy but positively with fertility [29].

**P1** relies on the high elasticities of infant and child survival, which are larger than elasticities to adult survival or fertility, to predict that recruitment of juveniles will be most important for population fitness differences [28]. However, if low-elasticity vital rates vary widely between populations, these rates may be more important drivers of population growth than high-elasticity rates (e.g., [30, 31]). Therefore, we systematically compare "importance" metrics to see how well prospective measures (elasticities) predict the vital rates that are actually driving population fitness difference (decomposed using retrospective LTREs).

Previous studies have pointed out limitations of elasticity analysis, such as differences in interpretation when comparing increasing vs. decreasing or small vs. large populations [32]. In addition, interventions based on elasticity analysis alter vital rates but also alter their elasticities [33]. In general, prospective (elasticity) analyses are useful for estimating the force of selection [23] and for identifying potential management targets [25], but LTREs are more appropriate for explaining observed differences in population performance [24].

**P2** is the intuitive prediction that population growth should reflect both survival and reproduction, since either will increase population growth, all else equal. However, longevity and fertility may trade off [34–36], so an increase in only one or the other may not increase population growth. For instance, greater life expectancy is associated with lower fertility across modern industrial nations [37], driving a negative correlation between life expectancy and population growth [38]. Therefore, the degree (and even the sign) of the correlations of population growth with fertility vs. longevity are empirical questions that we answer in the case of natural fertility subsistence populations and chimpanzees.

**P3** arises as a consequence of selection effects on the slow life history of primates [12], with slower life histories exhibiting higher elasticities in early vs. late life, both within and between species [39]. When infant mortality is low, more survive to maturity, thereby reducing the importance of recruitment on population growth. A longer reproductive lifespan also permits replacement of dead offspring with later births, while low fertility raises the average age of a population. Because all of these effects make early infant survival less important to population fitness, elasticity to child survival is predicted to correlate negatively with life expectancy but positively with fertility (P3, [40]). We extend this logic to predict that elasticity to infant survival should also correlate positively with the pace of fertility, and thus negatively with mean age at first birth (*AFB*), mean age of childbearing (*MAC*) and inter-birth intervals (*IBI*) since smaller values increase fertility, but positively with age at last birth (*ALB*).

## Materials and methods

### Demographic data

We examine published fertility and mortality rates estimated for ten contemporary, non-industrial small-scale societies with natural fertility and minimal to no access to modern medicine during the period of study (S1-S3 Tables in S1 Data; S1 Text in S1 Data contains ethnographic details): Australian Aborigines (Northern Territory, Australia), Ache (Paraguay), Agta (Philippines), Gainj (Papua New Guinea), Hadza (Tanzania), Herero (Namibia), Hiwi (Venezuela), Ju/'hoansi! Kung (Botswana and Namibia), Tsimane (Bolivia) and Yanomamo (Venezuela and Brazil). We also examine seven chimpanzee populations, including published data for five wild populations at Gombe and Mahale (Tanzania), Kanyawara and Ngogo (Uganda), and Taï (Ivory Coast), a captive population in the Taronga Zoo (Sydney, Australia), and a reintroduced (captive-founded but wild-breeding) population in Gambia (S1-S3 Tables in S1 Data; S1 Text in S1 Data contains metadata). These captive and managed populations are not included in species-level comparative statistics or composite life histories, but are used to reflect "best-case" scenarios for chimpanzees: low mortality in the protected and provisioned Gambia population and high fertility in the captive breeding program at Taronga Zoo. Because fertility estimates for Ngogo chimpanzees are not yet published, we estimate contributions applying fertility estimated at nearby Kanyawara. Also, because the Taronga Zoo mortality data includes few chimpanzee deaths we use mortality data averaged across three zoo populations [41].

We employ parametric models of mortality and non-parametric models of fertility to obtain smoothed annual rates (see S1 Text in S1 Data for details). Briefly, Siler's [42] five-parameter competing hazard model of mortality jointly models juvenile, age-independent and adult mortality. The Siler model, estimated here with a non-linear regression model (NLIN procedure in SAS 9.4), was employed in previous treatments of human subsistence and chimpanzee mortality because of its simplicity, robustness and interpretability of its parameters [17, 43, 44]. Using the statistical software *R* (version 3.5.1), we smooth raw fertility data with a local polynomial regression (loess; span = 0.5) and constrain the smoothed data to the observed

ages of reproduction by heavily weighting zero values in the single-year age-classes before the minimum age at first birth (age $\alpha$) and after the last recorded birth (age $\omega$), and imputing values outside this range as zero. Resulting smoothed fertility was rescaled evenly across age to conserve the *TFR* from raw data (S1 Fig in S1 Data) and statistical predictions of *AFB* and *ALB* are close to those of source estimates (S2 Fig in S1 Data).

## Data analysis

We construct a female age-structured Leslie [45] population projection matrix **A** ($\mathbf{A} = \{a_{ij}\}$) where matrix elements $a_{ij}$ describe the number ($n_i$) of age $i$ individuals alive in the population at time $t+1$ that are contributed by one age $j$ individual alive at time $t$, either via individual survival ($a_{x+1,x} = p_x$) or fertility transitions ($a_{1x} = m_x$) ([19]; Table 1 contains variable definitions; S1 Text in S1 Data contains details of matrix model methods and calculations of life history traits). Population size is updated by applying the population projection matrix **A** to the population age structure **n** ($\mathbf{n} = \{n_i\}$) and stable asymptotic population growth is described by the

**Table 1. Variable definitions.**

| Symbol | Variable | Equation | Symbol | Variable | Equation |
|---|---|---|---|---|---|
| $AFB$ | mean age at first birth | $AFB = \sum_x m_x \prod_{a=0}^{x-1}(1-m_a)$ | $MAC$ | mean age of childbirth | $MAC = \frac{2.05}{TFR} \sum_x x\, m_x.$ |
| $ALB$ | mean age at last birth | $ALB = \sum_x m_x \prod_{a=x}^{T}(1-m_a).$ | $m_x$ | fertility rate (daughters) | $m_x = a_{1x} = ASFR/2.05$ |
| $\mathbf{A}^{(n)}$ | population projection matrix | $\mathbf{A}^{(n)} = \{a_{ij}\}^{(n)}$ | $n$ | population index | $n = 1, 2, 3, \ldots, M$ |
| $\mathbf{A}^{(R)}$ | reference population | $\mathbf{A}^{(HG)} = E_n(\mathbf{A}^{(n \in \text{hunter gatherers})})$ | $n_{x,t}$ | population size age $x$ at time $t$ | $n_{x,t} = \mathbf{A}\, n_{x,t-1}$ |
| $a_{ij}$ | matrix element | age $i$ added at ($t+1$) per age $j$ alive ($t$) | $p_x$ | survival probability | $p_x = a_{x+1,x}$ |
| $C_a$ | adult survival effect (relative) | $C_c = \left(\sum_{x=0}^{\alpha-1} |C_{x+1,x}|\right) / \left(\sum_{0}^{T} |C_{x+1,x}| + \sum_{AFB}^{ALB} |C_{1x}|\right)$ | $q_x$ | probability of death | $q_x = 1 - p_x$ |
| $C_c$ | child survival effect (relative) | $C_f = \left(\sum_x |C_{1x}|\right) / \left(\sum_x |C_{x+1,x}| + \sum_x |C_{1x}|\right)$ | $r$ | intrinsic growth rate | $r = \log(\lambda)$ |
| $C_f$ | total fertility effect (relative) | $C_{ij} = s_{ij} \Delta a_{ij};\ \Delta\lambda = \sum_{i,j} C_{ij}$ | $s_{ij}$ | sensitivity | $s_{ij} = \Delta\lambda / \Delta a_{ij}$ |
| $C_{ij}$ | LTRE contribution (+/-) | $C_{ij*} = C_{ij} \div \left(\sum_{i,j} |C_{ij}|\right); 1 = \sum_{i,j} |C_{ij*}|$ | | total fertility rate | $TFR = \sum_x ASFR = \sum_x 2.05\, m_x.$ |
| $C_{ij*}$ | LTRE effect (relative magnitude) | $C_s = s_{x+1,x} \Delta a_{x+1,x}$ | **v** | reproductive value | left eigenvector of **A** |
| $C_s$ | survival contribution | $E_s = \sum_x e_{x+1,x};\ 1 = E_S + E_F.$ | **w** | stable age distribution | right eigenvector of **A** |
| $E_a$ | total elasticity to adult survival | $E_f = \sum_x e_{1x}$ | $T$ | maximum age at death | $T = \min(x | p_x = 0)$ |
| $E_c$ | total elasticity to child survival | $E_f = e_{1x}$ | $x$ | age | $x = \{0, 1, 2, \ldots, T\}$ |
| $E_f$ | total elasticity to fertility | $E_s = e_{x+1,x}$ | $Z_c$ | ratio of elasticities (child survival) | $Z_c = C_c : E_c$ |
| $E_s$ | total elasticity to survival | $E_0 = \max(e_{ij}) = e_{21}$ | $Z_f$ | fertility contribution: elasticity ratio | $Z_f = C_f : E_f$ |
| $E_0$ | maximum elasticity (newborn survival) | $e_0 = \sum_x l_x$ | $\alpha$ | minimum age at first birth | $\alpha = \min(x | m_x > 0)$ |
| $e_0$ | life expectancy (at birth) | $e_{ij} = (a_{ij} / \lambda)\, s_{ij}$ | $\lambda$ | population growth rate | dominant eigenvalue of **A** |
| $IBI$ | inter-birth interval | $IBI = TFR / (ALB - AFB + 1)$ | $\mu_x$ | mortality rate | $\mu_x = \log(1 - p_x)$ |
| $l_x$ | survivorship | $l_x = \prod_{a=0}^{x-1} p_a$ | $\omega$ | maximum age at last birth | $\omega = \max(x | m_x > 0)$ |

Columns contain the variable symbol, variable name and source equation for the demographic parameters estimated in our analyses.

dominant eigenvalue $\lambda$ ($\mathbf{n}(t+1) = \mathbf{A}\,\mathbf{n}(t) = \lambda\,\mathbf{n}(t)$). From the matrix $\mathbf{A}$ we calculate vital rate sensitivities ($s_{ij} = (\mathrm{d}\,\lambda\,/\,\mathrm{d}\,a_{ij})$) reflecting the force of selection on a vital rate as well as elasticities ($E_{ij}$) scaling the proportional effect on population growth ($E_{ij} = s_{ij}\,(a_{ij}\,/\,\lambda)$ [19]). Because elasticities conveniently sum to unity ($1 = \Sigma_{i,j}\,E_{ij}$), we can add elasticities across vital rates across age $x$ to estimate the total elasticity to survival ($E_s = \Sigma_x\,E_{x+1,x}$) or to fertility ($E_f = \Sigma_x\,E_{1x}$; $1 = E_s + E_f$), or sum across specific ages (e.g., before or after reproductive maturity at age $\alpha$) to distinguish the elasticity to survival through childhood ($E_c = \Sigma_{x<\alpha}\,E_{x+1,x}$) vs. elasticity to survival through adulthood ($E_a = \Sigma_{x\geq\alpha}\,E_{x+1,x}$; $E_s = E_c + E_a$).

Differences in population growth rates ($\lambda$, $r = \ln\lambda$) are decomposed into positive and negative contributions ($C_{ij}$) made by vital rate differences ($\Delta a_{ij}$) to the total difference $\Delta\lambda$ using a one-way fixed-treatment life table response experiment, or LTRE ($\Delta\lambda = \Sigma_{i,j}\,C_{ij} = \Sigma_{i,j}\,s_{ij}\,\Delta a_{ij}$; $\Delta a_{ij} = a_{ij}^{(m)} - a_{ij}^{(R)}$; [19]; S1 Text in S1 Data). Here, each population $m$ ($m = 1, 2, 3, \ldots, M$) is compared to a common (composite) reference (R) life history, here exhibiting the average fertility and survival rates estimated across hunter-gatherers (labeled HG and summarized in the matrix $\mathbf{A}^{(HG)}$). Species differences are highlighted by comparing this common reference to composite life histories exhibiting vital rates averaged across all wild chimpanzees (WC), and within-species differences are summarized by results for composite life histories estimated separately for exclusive hunter-gatherers (HG) vs. non-forager (NF) subsistence populations and for increasing (WC+) vs. decreasing chimpanzees (WC-). In addition to vital rate *contributions* ($C_{ij}$) that sum to estimate the total difference in population growth rates ($\Delta\lambda \approx \Sigma_{i,j}\,C_{ij}$), we also examine combined *effects* ($C_{ij^*} = |C_{ij}|\,/\,\Sigma_{i,j}\,|C_{ij}|$). Because these metrics are analogous to elasticities in that they sum to unity ($1 = \Sigma_{i,j}\,C_{ij^*}$), they reflect the proportion of total fitness contributions due to effects restricted to certain life stages (e.g., across childhood vs. adulthood: $C_c = \Sigma_{x<\alpha}\,C_{x+1,x^*}$; $C_a = \Sigma_{x\geq\alpha}\,C_{x+1,x^*}$; $C_s = \Sigma_x\,C_{x+1,x^*}$; $C_f = \Sigma_x\,C_{1x^*}$; $1 = C_s + C_f$; $C_s = C_c + C_a$). Therefore, we can examine the relative 'importance' of each life cycle component for driving population growth and we compare those metrics directly to prospective elasticities using ternary diagrams that predict the potential for fitness effects of fertility vs. child and adult survival [46]. For more detailed comparison, fertility is binned into early, prime and late fertility effects at the ages when completed fertility is 0–25%, 25–75% and 75–100% of the total fertility rate (*TFR*) in the hunter-gatherer reference (ages 0 to 22, 23 to 35, and 36 to 50, respectively). To aid interpretation of population differences, we calculate standard demographic rates: mortality hazard ($\mu_x$), survivorship ($l_x$), life expectancy at birth ($e_0$), total fertility rate (*TFR*), mean age at first birth (*AFB*), mean age of childbearing (*MAC*), mean age at last birth (*ALB*) and mean inter-birth intervals (*IBI*) (Table 1; S1 Text in S1 Data contains calculations).

We also evaluate three predictions of population biology and life history theory (**P1-P3**).

**P1.** Because early survival is under the strongest selection, reflected in high fitness elasticities for early survival, differences in early survival should have the largest effect on population growth. This is because fitness contributions ($C_{ij}$) of matrix element ($a_{ij}$) that reflect vital rate differences ($\Delta a_{ij}$) are scaled by elasticities $E_{ij}$ ($C_{ij} = E_{ij}\,\Delta a_{ij}$). Because early survival ($p_0$) has highest elasticity ($E_0$), it has the potential to make the largest fitness contributions, given the same proportional difference in a particular vital rate.

Child survival has the largest potential for fitness effects [28], so we expect child survival differences to have substantial effects on population performance. However, strong stabilizing selection may canalize important rates and reduce temporal variation within populations [26]; if such canalization applies broadly across environments, population variation in those age-specific vital rates may be limited as well, thereby reducing those LTRE contributions [27]. Our between-population comparisons are not necessarily a reliable 'space-for-time' substitution for vital rates under selection [47], but time-series demographic data exist for only a few study populations. Analyses of those longitudinal data revealed similar level of vital rate

variation within groups over time as between them [17]. Thus, if the variation in vital rates documented across continents, cultures and environments is similar to that observed within populations over time, we should expect child survival effects based on cross-population analysis to also be smaller than elasticities predict [27].

We calculate a scalar ratio ($Z_{ij}$) that reflects the actual realized fitness contributions of vital rates, relative to the potential suggested by elasticities ($Z_{ij} = C_{ij^*} / E_{ij}$). Because both vital rate effects ($C_{ij^*}$) and elasticities ($e_{ij}$) sum to unity, we can estimate $Z$ across all of childhood ($Z_c = C_c / E_c$) or across adulthood ($Z_a = C_a / E_a$), as well as for lifetime survival ($Z_s = C_s / E_s$) and for lifetime fertility ($Z_f = C_f / E_f$). Values of $Z > 1$ indicate greater importance of actual fitness contributions based on retrospective LTREs, whereas $Z < 1$ indicates contributions smaller than the potential indicated by prospective elasticities.

**P2.** We calculate correlations between population growth rates ($r = \ln \lambda$) and two emergent life history traits: life expectancy ($e_0$) and lifetime fertility ($TFR$).

**P3.** After confirming that early infant survival ($p_0 = a_{12}$) has the highest elasticity rate ($E_0 = E_{21} = \max(E_{ij})$), we calculate correlations between $E_0$ and: (a) longevity ($e_0$), (b) lifetime fertility ($TFR$), (c) mean age at first birth ($AFB$), (d) mean age at childbearing ($MAC$), (e) mean age at last birth ($ALB$) and (f) inter-birth intervals ($IBI$). We report $p$-values from non-parametric Mann-Whitney-Wilcoxon rank-sum tests used for all statistical tests of differences in means; for associations we report Pearson correlation coefficients $r$ and significance $p$-values. All results were computed using *Matlab*.

## Results

### Vital rates and elasticities

**Mortality.** Early infant mortality (age 0–1) is higher, on average, among hunter-gatherers than among chimpanzees (increasing or declining), while late infant mortality (age 1–2) is higher among hunter-gatherers than among increasing chimpanzee populations. However, at all other ages mortality rates are lower among hunter-gatherers than chimpanzees (Fig 1A; S3 Fig in S1 Data). Non-foragers have lower mean mortality than hunter-gatherers except between ages 53–64, where they are equivalent. Human life expectancies in our sample are more than twice those of wild chimpanzees ($e_0$; $p = 0.005$, Wilcoxon rank sum test) and are marginally higher among non-foragers than among hunter-gatherers ($p = 0.056$; Fig 2A; S4 Table in S1 Data). Captive and reintroduced populations are at the upper end of the wild chimpanzee range of longevity and several hunter-gatherer populations are at the lower end of the human range (Fig 2A).

After age 4, human mortality rates are lower than those in any chimpanzee population, but early infant mortality (age 0 to 1) is higher than the chimpanzee mean in five small-scale societies (the Agta, Hadza, Hiwi, Ju/'hoansi! Kung and Yanomamo). Late infant mortality (age 1 to 2) in our sample is lowest among the managed Gambia chimpanzees, and the lowest mortality between ages 2 and 4 is among wild Ngogo chimpanzees (S3 Fig in S1 Data). Humans are marginally more likely than chimpanzees to survive to their later age of reproductive maturity ($l_\alpha$; $p = 0.099$), but are significantly more likely to survive to the mean age of childbirth ($l_M$; $p = 0.040$) and to the maximum age of reproduction ($l_\omega$; $p = 0.005$; Fig 2A; S4 Table in S1 Data). These species differences are driven more by non-foragers ($p = 0.095$ [$l_\alpha$]; $p = 0.032$ [$l_M$], $p = 0.016$ [$l_\omega$]), since survivorship among hunter-gatherers is lower than among non-foragers ($p = 0.032$ [$l_\alpha$]; $p = 0.016$ [$l_M$]; $p = 0.095$ [$l_\omega$]). Only survivorship to the maximum $ALB$ is significantly higher among hunter-gatherers than among chimpanzees ($l_\omega$; $p = 0.032$; Fig 2A, S4 Table in S1 Data).

**Fertility.** Although mean survival-conditioned fertility ($TFR$) is similar among humans and chimpanzees ($p > 0.1$; Fig 2B; S4 Table in S1 Data), maximum lifetime fertility is highest

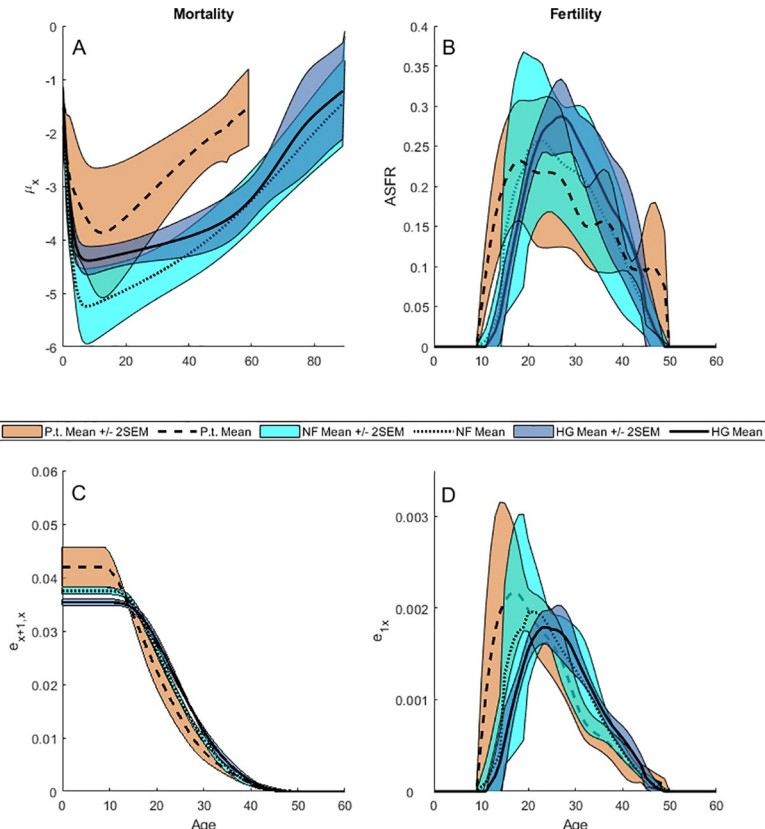

**Fig 1. Summary statistics for vital rates and elasticities.** 95% Confidence Intervals (Mean ± 2 SEM) are calculated across the age-specific vital rates estimated for five hunter-gatherer societies (dark blue fill, solid lines), five non-exclusive forager societies (light blue fill, dotted lines), and five wild chimpanzee populations (red fill, dashed lines). (A), Mortality ($\mu_x$). (B), Age-specific fertility rate (*ASFR*). (C), Survival elasticities ($E_{x+1,x}$). (D), Fertility elasticities ($E_{1x}$).

among humans (Tsimane *TFR* = 9.2; Fig 2B) and as noted above ($l_\omega$ comparison), very few chimpanzees survive to complete their potential *TFR* (10% of chimpanzees, compared to 33% of hunter-gatherers and 49% of non-foragers). As noted in previous studies [6], chimpanzees have earlier mean *AFB* (*p* = 0.001) and *MAC* (*p* = 0.005) than humans, but later mean *ALB* (*p* = 0.037) and longer *IBI*s (*p* = 0.017) (Fig 2B, S4 Table in S1 Data). Earlier *ALB* among managed chimpanzees is due to small sample sizes and use of contraception at Taronga Zoo [48] and other factors related to prior captivity at Gambia [49], so we include only wild chimpanzees in our correlations and difference tests. While both hunter-gatherers and non-foragers have later *AFB* than chimpanzees (*p* = 0.008 for each), non-foragers and chimpanzees have similar *MAC* (*p* = 0.056) and *IBI* (*p* > 0.1), and hunter-gatherers and chimpanzees have similar *ALB* (*p* > 0.1). Chimpanzee interbirth intervals calculated using these *AFB* and *ALB* estimates (mean±SD *IBI* = 3.6±0.3y) exceed human *IBI*s (*p* = 0.017), but this difference is only significant for hunter-gatherers (*p* = 0.008; S4 Table in S1 Data). These chimpanzee *IBI*s are also shorter than the 5.1–6.2y intervals reported elsewhere [6, 50]. As might be expected, our *IBI* estimate falls between those calculated for mothers whose offspring died before vs. after age four (2.2 y and 5.7 y, respectively [6]), with our lower estimate reflecting the averaged effects of infant mortality on birth spacing. Closer examination shows population differences in the tempo of fertility (Fig 2B; S3 Fig in S1 Data).

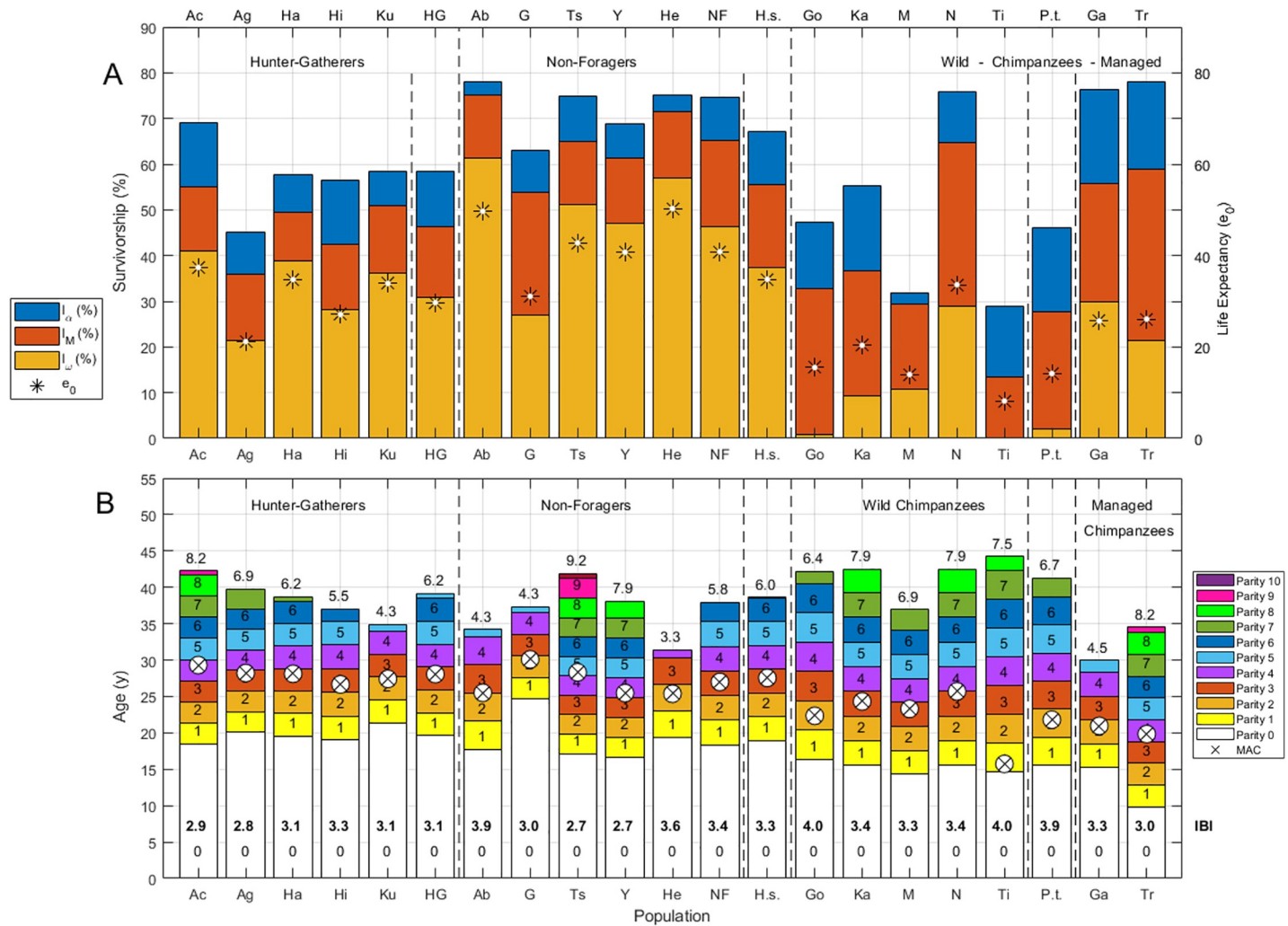

**Fig 2. Summary of demographic measures for chimpanzee and human populations.** (A) Stacked bars indicate survivorship (%) to maturity ($l_\alpha$), to mean age of reproduction ($l_M$) and to maximum age of reproduction ($l_\omega$); life expectancy ($e_0$) is indicated by a large asterisk. (B) Stacked bars indicate rough ages at each parity from zero (nulliparous) to the total fertility rate (*TFR*) estimated for each population. Parity is indicated by inset text in each stacked bar, *TFR* is indicated above each bar and the mean interbirth interval (*IBI*) is in bold text inset in the lowest (nulliparous) bar. Vertical dashed lines separate Hunter-gatherers (HG), Non-Foragers (NF) and Wild Chimpanzees (WC). Hunter-gatherers are labeled: Ac (Ache), Ag (Agta), Ha (Hadza), Hi (Hiwi), Ku (Ju/'hoansi! Kung), hunter-gatherer mean life history (HG). Non-foragers are labeled: Ab (Aborigines), G (Gainj), Ts (Tsimane), Y(Yanomamo), He (Herero), non-forager mean life history (NF). Chimpanzee populations are labeled: Go (Gombe), Ka (Kanyawara), N (Ngogo), Ti (Taï), wild chimpanzee mean life history (WC), Ga (Gambia), Tr (Taronga).

**Elasticities.** Compared to chimpanzees, human elasticity to early infant survival is lower ($E_0$; $p = 0.001$; $p = 0.008$ [hunter-gatherers]; $p = 0.016$ [non-foragers]), but elasticities to child survival ($E_c$; $p = 0.099$; $p = 0.095$ [hunter-gatherers]; $p > 0.1$ [non-foragers]) and to adult survival are similar to chimpanzees ($E_a$; $p > 0.1$), and total elasticity to human fertility is lower ($E_f$; $p = 0.001$ [humans]; $p = 0.008$ [hunter-gatherers]; $p = 0.016$ [non-foragers]) (Figs 1D and 3A; S4 Fig in S1 Data; S4 Table in S1 Data). Fertility elasticities may climb rapidly with age (e.g., Herero, Yanomamo and Taï chimpanzees) or slowly (e.g., Ache, Hadza and Tsimane) depending on the pace of fertility, but decrease at approximately the same rate as survival elasticities due to mortality attrition affecting both simultaneously (Fig 1C and 1D).

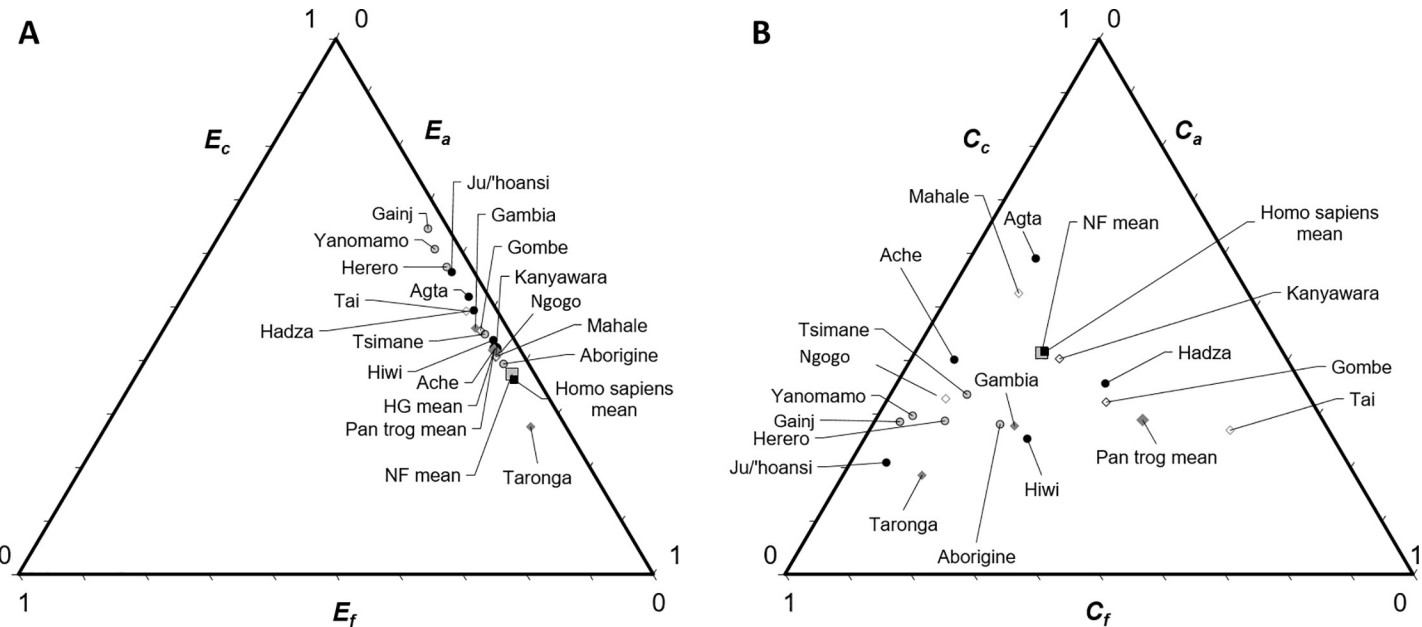

**Fig 3. Ternary diagrams of elasticities and contributions.** (A) Populations are arranged using the summed fitness elasticities for vital rates underlying child survival ($E_c$, left axis), adult survival ($E_a$, right axis) and fertility ($E_f$, bottom axis). (B) Populations are arranged using the summed fitness effects (contribution magnitudes) made by differences in child survival ($C_c$, left axis), adult survival ($C_a$, right axis) and fertility ($C_f$, bottom axis).

## Fitness contributions

All ten small-scale societies and two wild chimpanzee populations were growing, but two chimpanzee groups were declining slowly and one was collapsing (Figs 4 and 5). However, due

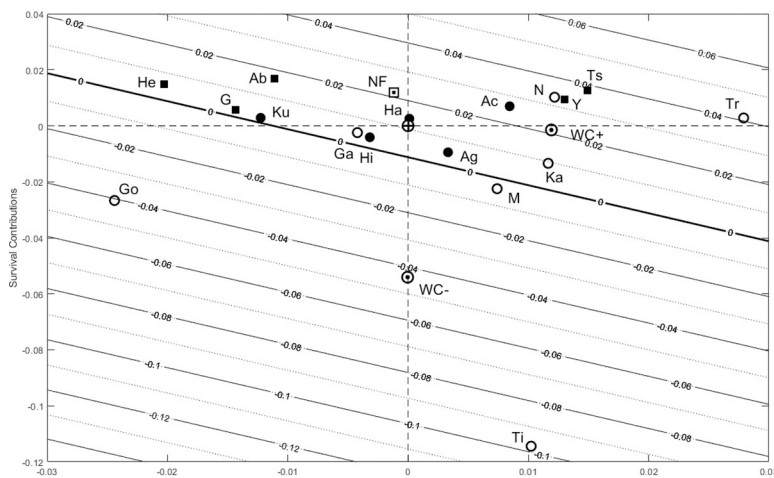

**Fig 4. Net contributions of fertility (*x*-axis) and survival (*y*-axis) from a Life Table Response Experiment (LTRE) comparing humans and chimpanzees to the mean life history estimated across five hunter-gatherer societies.** Hunter-gatherer societies are indicated by filled circles, non-foragers by filled squares and chimpanzees by open circles; the non-forager mean life history (labeled NF) and the mean life histories for declining (WC-) and increasing (WC+) chimpanzees are each indicated with a black-and-white dot, and the mean hunter-gatherer (HG) reference by a bullseye at the origin. Contours show population growth rate (*r*) isoclines with a bold line at *r* = 0. Compared to the HG reference, populations have positive net survival contributions if they fall above the horizontal dashed line and positive fertility contributions if they fall to the right of the vertical dashed line. Humans are labeled: Ab (Aborigines), Ac (Ache), Ag (Agta), G (Gainj), Ha (Hadza), He (Herero), Hi (Hiwi), Ku (Ju/'hoansi! Kung), Ts (Tsimane), Y (Yanomamo); chimpanzee populations are labeled: Ga (Gambia), Go (Gombe), Ka (Kanyawara), N (Ngogo), Ti (Taï), Tr (Taronga).

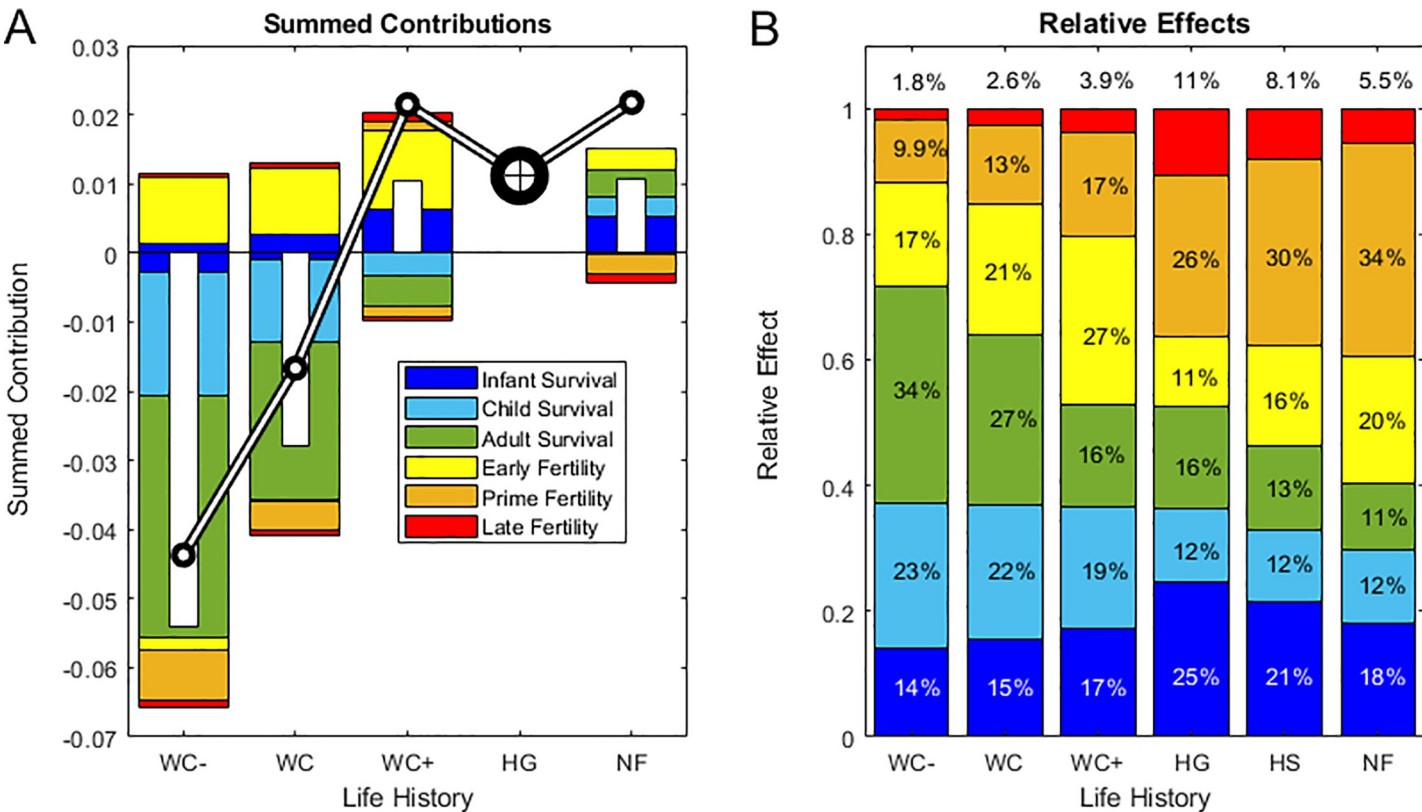

**Fig 5. Summed contributions for each composite life history.** (A), stacked bars show summed contributions of infant, child and adult survival and of early, prime and late fertility to the net difference ($\Delta r$) in population growth rate (inset white bars) between the composite mean hunter-gatherer reference (HG) and each focal population, with the black-and-white line crossing the bars indicating the focal population growth rate ($r = \log(\lambda)$). Note that positive and negative contributions are summed separately above and below the horizontal line at zero, and thus may reflect opposing contributions from the same life cycle component (e.g., negative and positive contributions of early vs. late infant mortality, respectively, if $\Delta p_0 < 0$ and $\Delta p_1 > 0$). Results are shown for four composite life histories with vital rates averaged over: declining chimpanzees (WC-), all wild chimpanzees (WC), increasing chimpanzees (WC+), hunter-gatherers (HG), or non-foragers (NF). HG (indicated by a bullseye) has zero contributions by definition because it is the common reference. (B), total effects ($\Sigma\ C_{ij^*}$) reflecting the combined magnitude of contributions, are averaged across the populations within each of the groupings in (A) plus averaged across all human groups (HS), in contrast to the results for the pre-averaged mean life histories shown in (A). Stacked bars decompose the mean total effect (the proportion of the combined magnitude of all contributions) made by infant, child and adult survival and by early, prime and late fertility (inset text shows the percent of total effects, with late fertility effects labeled above the bars).

to wide variation among our small sample, population growth differences were not statistically significant ($r = \log \lambda$, $p > 0.1$; S4 Table in S1 Data). Compared to the hunter-gatherer reference, declining wild chimpanzees had similar fertility but lower net survival contributions and increasing chimpanzees had survival comparable to hunter-gatherers but higher net fertility contributions, while non-foragers had higher survival but slightly lower net fertility contributions (Figs 4 and 5A; S4 Table in S1 Data). Lower early infant (age 0) mortality elevated population growth among both increasing and declining chimpanzees, but higher mortality at other ages made negative net contributions in every chimpanzee population except for Ngogo (Figs 4 and 5A).

Positive contributions of chimpanzees' higher early fertility up to age 22 (age 29 at Kanyawara) were partially offset by lower prime-age fertility between ages 23 and 35, which comprises half of the mean human *TFR* (Fig 5A), with higher survival allowing positive net contributions of late fertility among increasing but not decreasing chimpanzees. The rapid population growth of non-foragers was mainly due to higher survival at all ages, but offset by prime and late-age fertility, which was lower than in the hunter-gatherer reference (Fig 5A).

Despite differences in the signs of fertility and survival contributions, the relative magnitudes of vital rate effects were similar across populations ($p > 0.01$, Fig 5B; S4 Table in S1 Data). The only significant difference between hunter-gatherers, non-foragers and chimpanzees was that survival effects are stronger among chimpanzees than non-foragers ($C_s$; $p = 0.032$) and fertility effects were stronger among non-foragers than among chimpanzees ($C_f$; $p = 0.032$; Fig 5B; S4 Table in S1 Data).

Because so few chimpanzees survive to advanced ages, large differences in the potential for late-life fertility contributed little to population growth. Among humans, high survival drove population growth among non-foragers; among hunter-gatherers, lower early fertility effects were offset by higher prime- and late-age fertility (Fig 5A; S5, S7 Figs in S1 Data).

Among chimpanzees, population decline at Gombe was similar to the rate calculated for the mean chimpanzee life history, whereas decline at Mahale was slower despite high infant mortality because of higher adult survival and early fertility. Positive population growth at Kanyawara and Ngogo was due to lower mortality and higher prime fertility (S6, S7 Figs in S1 Data). At Taï, high juvenile and adult mortality drove precipitous decline ($r = -9.6\%$) despite low infant mortality and fertility near the chimpanzee mean. The managed population at Gambia was near-stationary with longevity balancing low fertility, and the Taronga Zoo population was growing rapidly with an active breeding program.

Although human fertility was mostly lower than chimpanzees, the populations with the highest growth rates (i.e., Tsimane, Yanomamo, and chimpanzees at Ngogo and Taronga Zoo) also had high fertility (S5, S7 Figs in S1 Data). The Ju/'hoansi! Kung, Gainj and Hiwi were all near stationary population growth–the Hiwi because of low infant survival and low fertility, whereas the Gainj and Ju/'hoansi balanced higher survival with lower fertility (Figs 4 and 5A; S5, S7 Figs in S1 Data). Among Herero pastoralists, high survival at all ages offset very low fertility. The Agta and Hadza both had low early fertility, but high infant mortality drove slower growth among the Agta despite higher prime and late fertility. Relatively rapid growth among the Northern Territory Aborigines was due to high survival offsetting low fertility at all ages, whereas the Ache grew faster due to high prime and late fertility. Very rapid growth was due to survival and early fertility among the Yanomamo and due to survival and late fertility among the Tsimane.

We now test our three predictions governing the role of elasticities and fitness contributions on shaping the life course of humans and chimpanzees.

**P1.** In agreement with P1, infant mortality rates have the largest elasticity and in many human populations high infant mortality substantially reduced population growth relative to the HG reference. Early infant survival made the largest fitness contribution ($C^* = p_0$) in four out of five hunter-gatherer societies, four out of five non-foragers and three out of five wild chimpanzee populations, with later infant survival ($p_1$) making the largest contribution among the Hiwi and the Yanomamo and among chimpanzees at Mahale and Ngogo. Also consistent with P1, the combined effects of infant and child survival across the life cycle were larger than adult survival effects ($C_c > C_a$; $p < 0.001$ [humans]; $p = 0.008$ [hunter-gatherers]; $p = 0.048$ [non-foragers]; $p = 0.008$ [chimpanzees]) and larger than fertility effects in chimpanzees ($C_c > C_f$; $p = 0.008$ [chimpanzees]) (Fig 5B, S5 Table in S1 Data).

However, fertility effects were unexpectedly larger than child survival effects in humans ($C_c < C_f$; $p < 0.001$ [humans]; $p = 0.008$ [hunter-gatherers]; $p = 0.008$ [non-foragers]; Fig 5B; S5 Table in S1 Data, S8 Fig in S1 Data). Across all populations pooled and across hunter-gatherers alone, fertility and total survival effects were equivalent ($C_s \approx C_f$; $p > 0.1$), but total survival effects were larger than fertility effects among chimpanzees ($C_s > C_f$; $p = 0.016$) and smaller among non-foragers ($C_s < C_f$; $p = 0.008$) (Fig 5B; S5 Table in S1 Data).

Elasticities estimate the force of selection and reflect the *potential* for fitness effects if vital rates differ, while the *observed* effects of vital rates depend on population-level differences. As

**Table 2. Fitness effect: Potential ratios.**

| Measure: | | $Z_c$ | $Z_a$ | $Z_s$ | $Z_f$ | Measure: | | $Z_c$ | $Z_a$ | $Z_s$ | $Z_f$ | Measure: | | $Z_c$ | $Z_a$ | $Z_s$ | $Z_f$ |
|---|---|---|---|---|---|---|---|---|---|---|---|---|---|---|---|---|---|
| Units: | | % | % | % | % | Units: | | % | % | % | % | Units: | | % | % | % | % |
| Ache | H | 95 | 13 | 49 | 1509 | **Aborigine** | A | 71 | 36 | 50 | 1318 | **Gombe** | W | 71 | 69 | 70 | 794 |
| Agta | H | 114 | 23 | 72 | 887 | **Gainj** | F | 44 | 12 | 34 | 2092 | **Kanyawara** | W | 99 | 42 | 66 | 894 |
| Hadza | H | 72 | 70 | 71 | 883 | **Tsimane** | F | 75 | 24 | 48 | 1450 | **Mahale** | W | 129 | 20 | 66 | 899 |
| Hiwi | H | 58 | 49 | 53 | 1338 | **Yanomamo** | F | 49 | 15 | 37 | 1599 | **Ngogo** | W† | 80 | 17 | 44 | 1411 |
| !Kung | H | 37 | 14 | 27 | 2083 | **Herero** | P | 50 | 29 | 41 | 1574 | **Tai** | W | 55 | 125 | 89 | 320 |
| **HG Mean** | * | 0 | 0 | 0 | 0 | **NF Mean** | * | 110 | 34 | 64 | 1028 | ***P.t.* Mean** | * | 69 | 79 | 74 | 684 |
| *H.s.* Mean | * | 115 | 34 | 64 | 1044 | * Mean Life History | | | | | | **Gambia** | M | 60 | 46 | 53 | 1079 |
| | | | | | | † Composite Life History | | | | | | **Taronga** | C† | 67 | 19 | 33 | 1249 |

Ratios ($Z = C/E$) of realized retrospective fitness contributions to the potential reflected in prospective elasticities for child survival ($Z_c$), adult survival ($Z_a$), all survival ($Z_s$) and fertility ($Z_f$). Separated rows show results for the mean life histories of hunter-gatherers (HG Mean, the LTRE reference), non-foragers (NF Mean), human small-scale societies (*Homo sapiens*, *H.s.* Mean), and wild chimpanzees (*Pan troglodytes*, *P.t.* Mean). Human subsistence modes in the second column are abbreviated: H (hunter-gatherer), A (acculturated hunter-gatherer), F (forager-horticulturalist) or P (pastoralist); chimpanzee management status is abbreviated: W (wild), M (managed) or C (captive).

these measures can differ widely (Fig 3A vs. 3B), the ratio of effect:potential is a useful metric to help inform us about the tradeoffs constraining life history evolution. Although elasticities predicted that child survival would contribute greatly to fitness differences, these rates did not account for as high a proportion of observed effects among humans as predicted on the basis of elasticities alone ($E_c$ vs. $C_c$, Fig 3). Adult survival effects were overestimated by elasticities (i.e., $Z << 1$ because $C << E$) even more than juvenile survival effects among chimpanzees ($Z_c > Z_a$; $p = 0.016$) and (marginally) more among non-foragers ($Z_c > Z_a$; $p = 0.095$). Across all populations, fertility effects were grossly underestimated by elasticities ($Z_c << Z_f$; $p = 0.008$; Fig 3; Table 2; S5 Table in S1 Data), hence all populations hover to the right of the elasticity ternary triangle (Fig 3A) but are more scattered in the fitness triangle (Fig 3B).

**P2.** Consistent with P2, population growth ($r$) was positively correlated with life expectancy ($e_0$) across our two-species sample ($r = 0.67$, $p = 0.006$) and (marginally) across chimpanzee populations ($r = 0.83$, $p = 0.084$), whereas population growth ($r$) and fertility ($TFR$) were positively correlated across human societies ($r = 0.81$, $p = 0.004$; $r = 0.95$, $p = 0.014$ [non-foragers]; $r = 0.82$, $p = 0.090$ [hunter-gatherers]; Table 3). Inconsistent with P2, population growth was

**Table 3. Life history correlations.**

| | Correlation | | All | *P.t.* | *H.s.* | HG | NF |
|---|---|---|---|---|---|---|---|
| **P2** | $r$ | $e_0$ | 0.67** | 0.83† | 0.42 | 0.62 | 0.08 |
| | $r$ | $TFR$ | 0.12 | 0.20 | 0.81** | 0.82† | 0.95* |
| **P3** | $E_0$ | $e_0$ | -0.41† | -0.57 | **0.69*** | 0.11 | 0.73 |
| | $E_0$ | $TFR$ | 0.30 | 0.08 | 0.12 | -0.40 | 0.28 |
| | $E_0$ | $AFB$ | -0.90*** | -0.39 | -0.83** | -0.30 | -0.93* |
| | $E_0$ | $MAC$ | -0.89*** | -0.95* | -0.87*** | -0.68 | -0.91* |
| | $E_0$ | $ALB$ | -0.06 | 0.55 | -0.29 | -0.41 | -0.16 |
| | $E_0$ | $IBI$ | 0.33 | 0.62 | 0.29 | **0.88†** | 0.20 |

Rows show correlations of: (P2) population growth rates ($r$) with life expectancy ($e_0$) or fertility ($TFR$); (P3) elasticity to child survival ($E_0$) with life history traits ($e_0$, $TFR$, $AFB$, $MAC$, $ALB$, $IBI$). Columns indicate Pearson coefficients for correlations across all populations pooled (All), across wild chimpanzee populations (*P.t.*), across small-scale human societies (*H.s.*), across hunter-gatherers (HG), or across non-foragers (NF). Significance is indicated by superscripts (*** $p < 0.001$, ** $p < 0.01$, * $p < 0.04$, † $p = 0.050$). Bold values indicate deviations from predictions (P3).

not correlated with life expectancy across humans or with fertility across chimpanzees ($p > 0.1$; Table 3).

**P3.** We expected that longevity should decrease, and fertility increase, the fitness elasticity to recruitment (corr($E_0$, $e_0$) < 0; corr($E_0$, $TFR$) > 0) [39]. Consistent with P3, elasticity to early infant survival is negatively correlated (marginally) with life expectancy across species (corr($E_0$, $e_0$); $r$ = -0.41, $p$ = 0.051; Table 3), but not across chimpanzees ($p > 0.1$). Inconsistent with P3, $E_0$ is *positively* correlated with $e_0$ across humans ($r$ = 0.69, $p$ = 0.029), and there is no correlation between $E_0$ and $TFR$ within or across species ($p > 0.1$; Table 3). Across our pooled two-species sample we find predicted (P3) negative correlations of $E_0$ with $AFB$ ($r$ = -0.90, $p < 0.001$) and $MAC$ ($r$ = -0.89, $p < 0.001$), but not with $ALB$ ($p > 0.1$). Among chimpanzees alone, only $MAC$ is negatively correlated with $E_0$ ($r$ = -0.95, $p$ = 0.014); among humans, $E_0$ is negatively correlated with $AFB$ ($r$ = -0.83, $p$ = 0.003; $r$ = -0.93, $p$ = 0.023 [non-foragers]; $p > 0.1$ [hunter-gatherers]) and $MAC$ ($r$ = -0.88, $p$ = 0.001; $r$ = -0.91, $p$ = 0.032 [non-foragers]; $p > 0.1$ [hunter-gatherers]; Table 3) but not with $ALB$ or $IBI$ ($p > 0.1$). Among hunter-gatherers there is a (marginal) *positive* correlation between $E_0$ and $IBI$ ($r$ = 0.81 $p$ = 0.050).

## Discussion

Although elasticities usually identify survival, especially juvenile survival, as the most important vital rate affecting fitness (**P1**, [28]) other vital rates may still have large effects. Juvenile survival was an important driver of population- and species-level differences (33% of all effects across human populations and 37% across chimpanzees), but adult survival was also an important driver (14% of all effects across humans and 27% among chimpanzees; Fig 5B; S8 Fig in S1 Data). However, fertility contributions were two orders of magnitude greater than expected based on the elasticities reflecting their potential, and fertility played a large role in regulating the five populations nearest stationarity (four out of five hunter-gatherer groups and one foraging-horticulturalist group): low fertility balanced longevity in four populations and high late-life fertility compensated for high infant mortality in one (the Agta). High fertility also drove rapid increase in the fastest-growing populations (Yanomamo, Ngogo, Tsimane and Taronga). That we found such large contributions of fertility differences (54% of all effects among humans and 36% among chimpanzees; Fig 5B; S8 Fig in S1 Data) highlights the potential for low-elasticity vital rates to have large effects on population fitness when they differ more than high-elasticity rates [24]. This counterintuitive result is what we would expect if stabilizing selection canalizes the vital rates deemed important based solely on their high elasticities [26, 27]. The effects of higher early fertility of non-foragers nearly balanced the higher prime and late fertility of hunter-gatherers, and among non-foragers these opposing fertility effects were larger than survival effects. This highlights a valuable feature of LTRE contributions, which allow us to identify the vital rates driving opposing fitness effects at different stages of the life course, even when their signs and magnitudes balance to yield small net contributions.

Several findings suggest potential constraints on the evolution of slower human life histories. Child survival among small-scale societies overlaps with rates documented for chimpanzees and child survival varies much more across populations than adult survival. Lower variation across populations in adult human mortality estimates may reflect greater buffering of exogenous mortality sources through derived human traits like food storage, widespread food sharing and ethnomedicine. Higher variation in chimpanzee mortality may reflect transient dynamics causing chimpanzee declines over the past century, due in part to human impacts such as poaching, habitat destruction and infectious outbreaks [51]. Despite strong stabilizing selection, child survival also varies over time more than adult survival among

humans [52] and among non-human primates [53], reflecting greater juvenile vulnerability to environmental effects. Because of quality-quantity trade-offs in which high fertility often comes at the expense of infant survival under natural fertility [54–57], low and variable infant survival may also reflect costs of reproduction borne more by offspring than adults [35, 55], and these tradeoffs may limit demographic buffering [26] through variance-reduction in this important vital rate. Another possibility is that selection may not be as strong as elasticities predict due to lower genetic variation in traits influencing infant survival [58] or there may be negative genetic correlations beyond the phenotypic correlations we examine here [59, 60]. Due to the requirement that the sum of elasticities for transitions going into an age class have to be equal to the sum across outgoing transitions [61], $E_0$ must equal the sum of fertility elasticities ($E_0 = E_f$), whereas there is no such constraint on fitness contributions. Additionally, higher infant mortality among some human societies may also reflect the costs of short inter-birth intervals and overlapping child dependence. These conspicuous features of human life histories combine elements of slow life histories (late maturity and low adult mortality) with elements of a faster life history (high infant mortality and short inter-birth intervals), which are made possible through adult production surpluses, resource transfers and multigenerational cooperation [8, 62]. Because these transfers alter vital rates directly and extend indirect fitness contributions beyond reproductive ages [62, 63], they may also alter elasticities [64], and their resulting effects appear in LTRE contributions only through the vital rates they affect. For instance, recruitment may suffer during resource shortages but indirect fitness contributions of production transfers buffer child mortality effects [65, 66]. In addition, if negative fitness effects of stochastic environments exceed costs of reproduction, then high fertility may bet-hedge against child mortality [67], resulting in higher long-term recruitment than a conservative "slow" life history strategy that buffers child mortality by reducing fertility [68]. Although we do not have sufficient long-term data to assess the effects of environmental or demographic stochasticity, previous work showed that temporal variation could be an important driver of population dynamics over evolutionary time (17).

As predicted (**P2**), population growth rates were positively correlated with longevity across our two-species sample, but they were not correlated with fertility. Within species, population growth is decoupled from longevity among small-scale subsistence societies and from fertility among chimpanzees. Among humans this reflects a slow life history and long post-reproductive lifespan, during which direct fitness contributions are zero even if individuals contribute to the fitness of living offspring indirectly through grandparenting [63, 65, 69] and other types of intergenerational resource transfers [8, 62, 66]. In contrast, high chimpanzee adult mortality decouples fitness from potential fertility because the potential contributions of higher late-life fertility are largely forfeited due to mortality attrition (only 10% of chimpanzees survive to attain their potential *TFR*, compared to 33% across hunter-gatherers and 49% across non-foragers).

Among chimpanzees, fertility contributions reflect recent high estimates of wild chimpanzee fertility (mean *TFR* = 7.3) based largely on a published compilation [6]. This survival-conditioned *TFR* is much greater than earlier estimates of 3.4 based on fewer populations and fewer births [8, 66, 69]. Those earlier studies under-estimated the mean age at last birth (*ALB* = 27.7 y vs. our estimate of 41.3 y) and may have also under-estimated mean age of first birth due to differences between dispersing and non-dispersing females [70]. To our knowledge, the finding that potential fertility in chimpanzees is comparable with human subsistence societies has not been widely appreciated, including the paper from which the fertility data originate [6]. It is possible that mortality selection reveals late fertility only among a robust subset of chimpanzees, which might suggest more variation in fecundity among chimpanzees than in humans. If, however, adult mortality rather than fertility limits chimpanzees'

reproductive potential, then human and chimpanzee life histories would be even more similar under conditions of low adult mortality. Because *IBI*s in chimpanzees are lower when infants die early, average *IBI* is affected by early life survival [50]. Unlike humans with overlapping dependents, lower juvenile mortality would further lengthen chimpanzee *IBI*s, and therefore reduce lifetime fertility. Our finding that the long-lived populations have shorter *IBI*s and higher *TFR* suggests that *IBI* differences among chimpanzees are more due to ecological conditions favoring both fertility and survival rather than tradeoffs between fertility and infant survival. Low mortality at Kanyawara, and especially at Ngogo, demonstrates the potential for rapid chimpanzee population growth, with vital rates that are similar to those of some hunter-gatherers [7, 16]. However, because the Ngogo mortality data covers a period of expansion after extirpating a neighboring group [71], these mortality rates may reflect a transient expansionary phase of rapid growth [17, 51] instead of a sustainable long-term life history like asymptotic analyses assume. At the far extreme, the captive zoo population illustrates the most favorable conditions, where chimpanzees have the reproductive potential to increase as rapidly as human subsistence populations. While phylogenetic analysis shows that human uniqueness stems from longevity and short birth spacing more than age at maturity [72], we find that age at maturity interacts with adult mortality to drive species life histories apart by limiting prime-age fertility contributions among chimpanzees.

As predicted (**P3**) by Jones [29], longevity marginally eases selection on recruitment across our two-species sample, but child survival is at a greater premium among long-lived human populations, which in our sample also exhibit high fertility and rapid population growth. It is likely that the correlations Jones [29] observed were due to cultural practices driving greater negative co-variation between fertility and mortality in Coale-Demeney model life tables than among subsistence societies (his small sample included examples with modern contraception driving low fertility and modern medicine driving low mortality). Although recruitment selection is not correlated with fertility in our sample, it is negatively correlated with fertility up to the mean age of childbearing, suggesting that the onset and peak of fertility moderate the fitness importance of recruitment more than fertility completion or birth spacing.

Also, the finding that the effect:potential ratio of fertility differences are much larger among humans than among chimpanzees highlights both the wide variation in human fertility and the effect of low chimpanzee survivorship, which limits prime-age and late fertility contributions. Finally, the positive correlation between hunter-gatherer recruitment elasticity and inter-birth intervals suggests that recruitment is more important when reproductive effort is low. Rather than confirming predictions that recruitment should be more important when reproductive effort is high, longer *IBI*s among hunter-gatherers puts a premium on infant survival, whereas short *IBI*s reduce the importance of recruitment because they allow quicker replacement of lost offspring.

## Study limitations

Our sample is the largest to date for human subsistence populations and wild chimpanzees, but these populations in their recent environments may not accurately represent ancestral life histories. The circumstances surrounding subsistence lifestyles and resource ecology vary by geography, history of interactions with neighboring populations, governmental intervention and regulation of territory, as well as other factors. Although contemporary hunter-gatherers do not replicate ancestral demography even in earliest "pre-contact" periods, they are the best reflection of vital rates in the absence of modern amenities, and of the evolutionary context within which our species evolved [73]. These populations exhibit characteristics common across prehistory among small-scale societies, including natural fertility, non-market livelihoods, greater pathogen burden, and multi-generational cooperation.

Though imperfect representatives of the past, the differences we observe within and between species nonetheless offer a unique opportunity to learn about the forces shaping human history. Previous findings suggest that alternative demographic routes to human persistence are reflected in life history adaptations that maintain the potential for high intrinsic growth rates and allow recovery from periodic population crashes [17]. The Ju/'hoansi! Kung, Hiwi and Gainj, with low and delayed fertility, hover near-stationarity and are on the slower side of a life history continuum, while the Tsimane and Yanomamo are on the faster side with early and high fertility driving rapid population growth, perhaps in response to post-colonization recovery. The Hadza life history is very close to the hunter-gatherer composite reference, suggesting perhaps that they may represent a "typical" contemporary hunter-gatherer population. While the! Kung belong to the most ancient (L1) human haplogroup [74], their lower population growth may reflect habitat degradation, displacement by pastoralists, and secondary sterility from infection [75, 76]. Similarly, data on extant chimpanzees reflect novel anthropogenic influences but provide the best representation of the demography of ancestral hominins [77], with captive and managed populations providing additional insights about best-case scenarios. As with any pair of lineages, we are confronted with questions about the conditions under which human and chimpanzee life histories diverged, since they may have faced very different selection pressures over evolutionary time since their divergence from a common ancestor. Also, because we are sampling human societies that survived contact, our sample may over-represent growing populations, especially since these short-term data may have captured transient growth periods in population cycles with rapid or catastrophic declines and prolonged recovery [17].

## Conclusions

Since divergence from chimpanzee-like ancestors, human survival has increased so much that adult mortality profiles of pre-industrial human and chimpanzee barely overlap. While species differences in adult mortality have been widely recognized [21], we report additional species differences and similarities: hunter-gatherers have similar, and sometimes higher, infant mortality than chimpanzees, whereas fertility is much more variable across human societies and overlaps the range of chimpanzees, especially across prime childbearing years. However, due to high mortality attrition, the force of selection on chimpanzee fertility is much lower than for humans, and more strongly favors younger mothers.

Our findings suggest that the trajectory forward from the life history of our most recent common ancestor with the chimpanzee was likely not a monotonic decline in mortality and that high and variable infant mortality likely played a large role in regulating population growth over evolutionary time. We also find that fertility differences have substantial effects on population growth despite low elasticities, and that older reproductive individuals may contribute more to population-level fitness differences than younger individuals with higher reproductive values. The diverse environments humans inhabit are partly responsible for observed variation in reproductive success across populations, but quality-quantity tradeoffs between fertility and juvenile survival, combined with prolonged juvenile susceptibility, may constrain evolution of slower human life histories in subsistence societies with natural fertility. Because delayed fertility reduces selection on recruitment across species and among humans, this suggests a fast-slow continuum of life history even among extant hominins, with early *AFB* and strong recruitment selection on the fast side, and late *AFB* and weaker recruitment selection on the slow side. High and variable juvenile mortality reflects bet-hedging costs of reproduction, maintaining a high selective premium on juvenile survival even in longer-lived human populations. We also find that late-life fertility is an important driver of population-level differences among small-scale societies despite typically low survival to these ages, and

that longevity can maintain stationary populations despite low fertility. Age-patterns of mortality strongly influence the effects of fertility differences, with adult mortality, age at maturity and menopause driving human and chimpanzee life histories apart despite similar survival-conditioned fertility.

## Supporting information

**S1 Data.**
(DOCX)

## Acknowledgments

We thank Shripad Tuljapurkar, Oskar Burger, Thomas S. Kraft and two anonymous reviewers for helpful feedback on previous versions of this manuscript, and we thank Thomas S. Kraft for help smoothing the fertility data we employ.

## Author Contributions

**Conceptualization:** Raziel J. Davison, Michael D. Gurven.

**Data curation:** Raziel J. Davison, Michael D. Gurven.

**Formal analysis:** Raziel J. Davison.

**Investigation:** Raziel J. Davison.

**Methodology:** Raziel J. Davison.

**Project administration:** Michael D. Gurven.

**Software:** Raziel J. Davison.

**Supervision:** Michael D. Gurven.

**Visualization:** Raziel J. Davison.

**Writing – original draft:** Raziel J. Davison.

**Writing – review & editing:** Raziel J. Davison, Michael D. Gurven.

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
