## [Decision Letter · Decision Letter 0]

20 Oct 2020

PONE-D-20-26560

Human uniqueness illustrated by life history diversity among small-scale societies and chimpanzees

PLOS ONE

Dear Dr. Davison,

Thank you for submitting your manuscript to PLOS ONE. After careful consideration, we feel that it has merit but does not fully meet PLOS ONE’s publication criteria as it currently stands. Therefore, we invite you to submit a revised version of the manuscript that addresses the points raised during the review process.

We received two sets of reviewers’ comments (please note one is provided in a separate file only). I found both comments to be very constructive for improving the manuscript. Overall, the two reviewers think the study is interesting (or potentially interesting) but identified some technical problems. Although these are major problems, I think they can be resolved. Some of the comments from reviewer 2 revolve around the issue of linking mathematical quantities to evolutionary processes; this is a difficult issue. It is important to clarify the links by providing clear logic behind them.

We look forward to receiving your revised manuscript.

Kind regards,

Masami Fujiwara, PhD

Academic Editor

PLOS ONE

Journal Requirements:

2. In addition, please provide additional details regarding the data sources for chimpanzee data. Specifically, please state whether the data was directly collected by the researchers for the purposes or whether data was extracted from the the published literature. If chimpanzee colonies were monitored for the purposes of the study, please confirm whether permits were obtained from the local authorities. If permits where not required please state this within your Material and Methods. In addition, please ensure that you upload Supplemental File Table 1 containing the demographic information for the human populations included in this study.

3.We note that you have stated that you will provide repository information for your data at acceptance. Should your manuscript be accepted for publication, we will hold it until you provide the relevant accession numbers or DOIs necessary to access your data. If you wish to make changes to your Data Availability statement, please describe these changes in your cover letter and we will update your Data Availability statement to reflect the information you provide.

4. Please include a copy of Table 3 which you refer to in your text on page 32.

Reviewers' comments:

Reviewer's Responses to Questions

**Comments to the Author**

1. Is the manuscript technically sound, and do the data support the conclusions?

Reviewer #1: Partly

Reviewer #2: Partly

2. Has the statistical analysis been performed appropriately and rigorously? 

Reviewer #1: I Don't Know

Reviewer #2: No

3. Have the authors made all data underlying the findings in their manuscript fully available?

Reviewer #1: Yes

Reviewer #2: Yes

4. Is the manuscript presented in an intelligible fashion and written in standard English?

Reviewer #1: Yes

Reviewer #2: Yes

5. Review Comments to the Author

Reviewer #1: Please see the attachment of my review.

Please see the attachment of my review.

Please see the attachment of my review.

Please see the attachment of my review.

Please see the attachment of my review.

Reviewer #2: See attached pdf file.

Review of Human uniqueness illustrated by life history diversity among small-scale societies and chimpanzees by Raziel Davison and Michael Gurven.

In this article, the authors compare age-trajectories of survival and fertility of many hunter-gatherer and forager human populations and chimpanzees. They then perform LTRE and spectral analysis to investigate potential evolutionary changes between these trajectories. This is a very interesting manuscript, revisiting with originality and up-to-date data a classic question in human evolutionary biology: the evolution of the human and chimpanzee life-cycle since divergence. The manuscript incorporates a wonderful comparison of age-trajectories of the different not-industralized populations with several chimpanzee populations, again using the best data to date. In this respect I think that this article has great potential. I am more sceptical about the evolutionary interpretation of the LTRE and elasticity analysis. I think that there are several conceptual and technical issues which need to be addressed prior to publication.

Main comments 1

The authors mainly use two metrics in the analyses: elasticities and what they called fitness contributions. All over the manuscript, I found unclear the definition of fitness contribution, and what the two measures together brings to the analyse.

First, I think there is a problem of definition throughout the manuscript. This starts L 52-55 where I found the sentence “These fitness contributions illustrate how life history event schedules drive differences within and between species, while elasticities reflect the force of selection and highlight the potential for fitness contributions if vital rates vary across populations [19]” not very clear (illustrate? drive difference of what on what?). The authors quote [19] which focuses (to my knowledge) exclusively on elasticity and tells anything (as far as I remember it) on contrasting elasticity and ‘fitness contribution’.

The authors are later more explicit when referring to LTRE where they are defined as the “vital rate contributions to observed differences in population growth rates” between two projection matrices (please note that vital rates are not individual measures as mentioned in l71 since there are population aggregates). In this sense, they are not “contribution to [a population] fitness” but how differences in entries of two matrices translate into difference in change of population reference growth rate. I strongly suggest the authors to define it more clearly. A way to do it is that sensitivity sij is the impact on λ of one unit of change in matrix entry aij. If we multiply sij by Δaij, it tells us how such a change would have modify the reference population growth λ.

Second, the authors then states l74-75 that “Differences between realized fitness contributions and the potential suggested by elasticities may indicate constraints on life history evolution” (also 405-406). This can be a fantastic idea and I can intuit what the authors have in mind. Yet it is not trivial to me, and it makes me wonder if this has been already theorized elsewhere. If it has, the author should clearly state it and explain why (I think not shying away equations). If it has not, I would strongly encourage the authors to develop - and if possible demonstrate - this idea. For instance, Cij, is a given amount of change between two matrix waited by sensitivities. Does this idea relates to the long lasting debate on the difference between using sensitivities and elasticities?

Main comment 2

The authors used the ratio between contribution and elasticity to measure (if I understand well) these possible constraints. But, I would strongly suggest the authors to check the resulting equation. First, l146, I think there is a mistake: eij is not equal to sij*(λ/aij) but to sij*(aij/λ) (I guess that this is a typo because elasticities look ok in fig 1) .

But then Zij = Cij/Eij = (Δaij.sij)/(sij(aij/λ))=(Δaij/ aij)(1/λ).

Then Z is the ‘percentage’ of difference between the reference and the analysed matrices divided by the growth rate. I am far on being clear on what does this mean and how this allows identifying constraint on a vital rate. I therefore strongly suggest the authors to explicit this metric and how/why it is used to solve their research question.

I am also not clear on whether Z should be sum(Cij)/sum(Eij) or rather sum(Cij/Eij), which can be substantially different.

Finally I don’ t understand the values for the Zs in table 1. For instance for Ache, Zc=Cc/Ec=7/42=0.16, not 95. Or, am I missing something?

I would suggest to incorporate Table S5 into the main text.

Main comment 3

I am not sure that I understand prediction 1 and it may be there a conceptual mistake. Canalization is the fact that vital rates impacting the more fitness (here λ) should exhibit lower temporal variance than those under weaker selection. The authors rightfully quote [22] and [23] testing this by somehow correlating the estimation of the variation in time of matrix entries to the variation of λ (but variance is in time, not between populations, isn’t it?).

Note also that, if I am not mistaken, [23] performed elasticity analysis not LTRE (as suggested in sentence l84) such that the effect of variance on LTRE is also not that clear to me. Anyway, I cannot see how LTRE between populations (without temporal variance accounted for) can allow identifying life-history constraint and how the concept of canalization is involved into this. If I am mistaken, I strongly suggest the authors to make their point more clear.

Main comment 4

I find that P1 (l81-82) is not well formulated. If I am not mistaken, it is a property of elasticity to be strictly declining with age in an age-structured model, infant and children survival elasticity always being constant and the largest. Metric have to be twisted and parameters very different that those of mammals to find alternative pattern (Baudish, 2005, PNAS). It is between species that relative magnitude of elasticities can be compared and I would strongly suggest to cite Heppell et al., 2000, Ecology for a comparison in mammals across the slow-fast continuum. Also why not refering to and using a classical Silvertown triangle to represented this (Silvertown, J., et al. 1993. Jouranl of Ecology 81:465–476)?

I am not sure what the authors want to test with prediction 2 which is the obvious fact that both increasing survival and fertility should increase population growth rate. Evidencing trade-off between fertility and survival?

Main comment 5

I would suggest the authors to discuss limitations of elasticities analyses in general and apply to humans in particular. (1) First elasticities are only one hand of the evolutionary GxE equation (Lande 1982; Charlesworth 1990; Steppan et al. 2002). Evolution also need genetic variance and this could be acknowledged. (2) The authors are comparing Leslie matrix, but any sub structuring (as individual heterogeneity) or hidden trade-offs may change the results. (3) It the most important, it has been shown that intergenerational transfers between age-class or parental investment can strongly impacts elasticities on survival and fertility in humans (Lee 2003, PNAS, Pavard et al. 2007, Evolution, Pavard & Branger 2012 Theo Pop Biol). For instance, magnitude of elasticities on adult survival may be strongly underestimated when maternal or grand-maternal care is not implemented. Elasticities on fertilities by age can also exhibit very different patterns. Because such intergenerational transfers have been proposed as a very important drivers of the evolution of human life-history, the authors should at least discuss it. (4) As the authors wonderfully argued in a recent article, only periodic catastrophes in humans can explain the human forager paradox. It also means that all elasticity analysis in constant and infinite environment is somehow incomplete and elasticity should be considered into a stochastic model.

Minor comments & Détails

l418-420 – Isn’t there a contradiction is stating that juvenile survival is under canalization effect and stating later on that it varies more in time than adult survival?

Figure 1 – I am not sure how the SEM of elasticity is calculated. Is this trivial?

I am not sure what figure 2 really brings to the article since it is complicated and barely discussed.

Problem in legend of figure 2 – Non-Forager are filled-circled as HG not filled square. Indicate that isolines are population growth rates. Remove the title.

L290 – I guess this is fig3.B instead of 2B,3?

l 32 – I am not sure that reference [2] did anything in calculating the divergence time between humans and chimpanzees. Please check carefully this reference. I think it should instead be referred to l 33-34.

l 35, “human fertility is similar to chimpanzees” and further. Please be more specific. Do you mean the shape of the age-specific fertilities? If yes, both the distribution and the TFR? Is the whole shape the distribution identical? The authors refer to [6] who focus mainly on reproductive senescence and show that if the timing of reproductive senescence is similar rate of reproductive senescence is not the same as well as how it correlates with decline in survival. I suggest to be more precise.

L37-38 – “However, there is great variation among human and chimpanzee life histories”. Here again I suggest to be more specific. The authors quote [8]. Although a valid reference, it can be completed by more recent article (as the [2]). Furthermore, I am not a native speaker but is “difference” would be better than “variation”?

L39 and many time after – Please change “within species” by between population. In ecology, within species study refers more to the study of variance between individuals than between population as it is investigate here.

L 40 - “We interpret population life histories in terms of the slow-fast life history continuum [9]” – Why? Also, human a complete outlier on this continuum so that I wonder if this is relevant.

L41 – To my knowledge Stearns’ book (but I don’t have it at hand here to check) is about trade-offs in general not about their importance for human life-history evolution.

In [6] the authors use extensive data in chimpanzees. Yet, this represent only about 600-1000 individuals (the equivalent of a small human village) spread in small groups over nearly a continent. How this could affect the authors’ results?

L55 – I am not found of the concept of population fitness underlying in this sentence.

L71 – Indicates the pages in [15]. Note that you could have also quoted Hal Caswell, 1989, Analysis of life table response experiments I. Decomposition of effects on population growth rate, Ecological Modelling, Volume 46, Issues 3–4.

L153 – this should be sij instead of sj isn’t it?

L179 – I am not sure to understand why the fact that Cij and Eij sum to unity allow to calculate the ratio.

Figure 3A – I find the figure very complicated to figure out. Are they the mean summed C values between populations? Then why and how is separated positive and negative C values? Or “composite” refers to the mean trajectories for HG, F, WC. But then, again, how does it lead to both (+) and (-) for a same trait (i.e., Infant survival). I am very sorry if I miss this information.

6. PLOS authors have the option to publish the peer review history of their article (what does this mean?). If published, this will include your full peer review and any attached files.

Reviewer #1: No

Reviewer #2: No

---

## [Author Response · Author response to Decision Letter 0]

1 Jan 2021

RESPONSE TO REVIEWERS

Reviewer Comments for PONE-D-20-26560

Human uniqueness illustrated by life history diversity among small-scale societies and chimpanzees

Authors of this manuscript showed that delayed maturity and adult mortal-

ity is the main difference to separate humans from chimpanzees which shares

common ancestor. They employed Life Table Response Experiments to quantify

vital rate contributions to population growth rate. Their results and discussion

are is interesting. Their approach is justified. However, this manuscript con-

tains numerous inconsistencies in its data and statements. It requires careful

attention on details to justify its results and conclusions.

We thank you for the positive feedback. We have revised the paper considerably to account for all of the reviewers’ careful comments. Our responses are in bold.

Major inconsistencies and questions:

• Lines 213 - 227, Table 1: In the manuscript, line 180 states that Za =

Ca=Ea and lines 148 and 150 shows that Ec + Ea + Ef = 1. However,

numbers in the table are inconsistent to the statement in text. Za 6=

Ca=Ea and Ec + Ea + Ef 6= 1

We apologize for the typo in which the column headers for Cc and Ca were switched. [Line 289]

We have now made sure all notation is consistent throughout the paper: 

(Ec + Ea) + Ef = Es + Ef = 1 in every row, and in every row Z = C/E (e.g. Zf = Cf/Ef).

• Lines 213 - 227, Table 1: Why E0 = Ef ?

This is a fact that is demonstrated by loop elasticities, which show that the sum of elasticities coming into a life history stage must be equal to the sum of elasticities across all outgoing transitions. Here, E0 = E(a21) = sum(Efert) because the elasticity to recruitment (the only pathway out of the newborn “stage”) is equal to the sum of elasticities to fertility at different ages (Van Groenendael et al. 1994).

Van Groenendael, Jan, Hans De Kroon, Susan Kalisz, and Shripad Tuljapurkar. "Loop analysis: evaluating life history pathways in population projection matrices." Ecology 75, no. 8 (1994): 2410-2415.

• Lines 213 - 227, Table 1: Why do managed and captive population have

smaller ALB compare to Wild population for chimpanzees?

For the captive breeding program at Taronga Zoo, ALB estimates come from only 7 females, including one that died 3.5 y after the last birth, one on contraception, and one that was transferred (Littleton 2005), so this is likely an underestimate of captive ALB. In the Gambia population, they do not use contraceptives and AFBs are similar to wild chimpanzees, but there may be factors stemming from prior captivity that limit their reproductive lifespans (Marsden et al. 2006). These are some of the reasons that only wild chimpanzees are considered in the ALB comparisons (S2 table).

• Fig 3A: Why are there two yellow and red sections in the bar of WC- and

WC? Why are there two red and brown sections in the bar of WC+? Why

does Infant survival (the purple section) have both positive and negative

contributions in WC- and WC?

The bars above the y-axis origin (C = 0) show positive contributions and those below the origin show negative contributions. They sum together to the net contributions (shown as white bars), which sum to the total difference in population growth rate between the target (m) and Reference (R). This is clarified in the figure caption. [lines 370-373].

We consider infancy as from birth (age 0) to 2 yrs. Thus, there are two age-contributions of infant survival (p0 = a21, p1 = a32) that may be of opposing sign. For instance, with WC, newborn survival p0 (a21) makes positive contributions because it is higher than that of hunter-gatherers (R), but older infant survival p1 (a32) makes a negative contribution because age 1-2 survival estimates for the WC average and WC- (declining chimpanzee average) are lower than those of the hunter-gatherer average (R).

• Lines 337-338: \\E0 > Ec > Ea > Ef " is not consistent with numbers in Table 1.

We regret the confusion caused by these inequalities and we have simplified our explanation of P1, removing the inequality and explaining the prediction in clearer English. [line 416]

E0 (the elasticity of newborn survival p0, which could also be written as E21, the elasticity of matrix element a21) is the largest elasticity for a single transition (here, individual matrix elements). Ex+1,x<α is larger than Ex+1,x≥α at all ages and Ex+1,x > E1x except for the last couple years of reproductive life. However, the values in Table 1 are the sums across all ages (e.g. Ec = Σx<α Ex+1,x, Ea = Σx≥α Ex+1,x), so Ea may be larger than Ec because it sums over a larger range of ages (all ages after α, the minimum AFB). As stated above, E0 = Ef = Σx E1x , requiring that the elasticities to fertility at single ages are each lower than E0.

• Line 345: \\Cc > Ca" is not consistent with numbers in Table 1.

We have fixed a typo where the column headers for Cc and Ca were switched in Table 1. [lines 289, 423]

• Line 350: \\Cs _ Cf " is not consistent with numbers in Table 1.

This “≈” indicates there is no significant difference between Cs and Cf (p > 0.1), not that these values are equal. [line 429]

• Lines 357-359: \\Zc _ Za", \\Zc < Za" and \\Zc < Za" are not consistent with numbers in Table 1.

The inequalities were reversed and are now correct and in agreement with Table 1: [lines 437-440]

Zc ≈ Za (HG, p > 0.1), Zc > Za (WC, p = 0.016), Zc > Za (NF, p = 0.095), Zc << Zf (p ≤ 0.008).

We now clarify that, by saying that adult survival effects are more under-estimated than child survival effects, this means that Za if farther below 1:1 than Zc (so Zc > Za). 

Minor points:

• Line 127: please spell out \\NLIN".

Done [line 165]

• Lines 131 to 133: Is it a duplicated statement?

This statement is not duplicated elsewhere in the paper.

• Line 146: \\(eij = (@_=@aij) = sij(_=aij)" should be \\(eij = (@_=@aij) =

sij(aij=_))".

Typo fixed [line 186]. We also capitalize all elasticities (e.g. Eij) to avoid confusion with the lower-case e0 for life expectancy [lines 185-191].

• Line 153: \\ sumi;j sj_aij" should be \\ sum_i;j (sij_aij).

Typo fixed [line 197]

• Line 184: \\maxi;j(eij)" should be `max(eij)".

We now clarify that E0 (distinguished from life expectancy e0 by capitalization) is the highest elasticity (E0 = max(Eij)) – elasticity to age 0 survival in the matrix element a21. [lines 245-246]

• Table 1 row 1: Please add description of l_, lM, l!, TFR, AFB, MAC,

ALB and IBI in the caption.

Done [lines 274-277]. We also include a new Table 1 with variable definitions and source equations [lines 192-194, moved from Supporting Information].

• Table 1 row 1: Please switch column Ca and Cc for display consistency.

Done [line 289]

• Line 278, Fig 2: Cannot _nd the \\non-foragers by _lled squares" in Fig 2.

Not sure why. They were in the original file and we have made sure they are in the current file. [lines 347-349]

• Line 279, Fig 2: Cannot _nd the \\(labeled NF)" in Fig 2.

Again, not sure why. It is the unfilled square with a dot in it, located at (-0.001157,0.01994). 

• Line 349: \\S4 Table" should be "S2 Table".

Fixed to say S3 Table (Differences within populations) [line 425]

• Line 352: \\S4 Table" should be "S2 Table".

Fixed to say S3 Table (Differences within populations) [line 430]

• Line 360: \\S4 Table" should be "S2 Table".

Fixed to say S3 Table (Differences within populations) [line 440]

• It will be helpful to add a data table of computed age specific mortality

and fertility of each population (which are the data used to plot Fig 1)

into online Supporting Information (S2, S3 Tables).

Added [lines 880-885]

 

Review of Human uniqueness illustrated by life history diversity among small-scale societies and chimpanzees by Raziel Davison and Michael Gurven.

In this article, the authors compare age-trajectories of survival and fertility of many hunter-gatherer and

forager human populations and chimpanzees. They then perform LTRE and spectral analysis to investigate potential evolutionary changes between these trajectories. This is a very interesting manuscript, revisiting with originality and up-to-date data a classic question in human evolutionary biology: the evolution of the human and chimpanzee life-cycle since divergence. The manuscript incorporates a wonderful comparison of age-trajectories of the different not-industralized populations with several chimpanzee populations, again using the best data to date. In this respect I think that this article has great potential. I am more skeptical about the evolutionary interpretation of the LTRE and elasticity analysis. I think that there are several conceptual and technical issues which need to be addressed prior to publication.

We appreciate the positive comments. Below we address the reviewer’s concerns about conceptual and technical issues.

Main comments 1

The authors mainly use two metrics in the analyses: elasticities and what they called fitness contributions. All over the manuscript, I found unclear the definition of fitness contribution, and what the two measures together brings to the analyse. 

First, I think there is a problem of definition throughout the manuscript. This starts L 52-55 where I found the sentence “These fitness contributions illustrate how life history event schedules drive differences within and between species, while elasticities reflect the force of selection and highlight the potential for fitness contributions if vital rates vary across populations [19]” not very clear (illustrate? drive difference of what on what?). The authors quote [19] which focuses (to my knowledge) exclusively on elasticity and tells anything (as far as I remember it) on contrasting elasticity and ‘fitness contribution’. 

We now more carefully define “fitness contribution” [lines 88-95, 195-200], and clarify the distinction between this measure and fitness elasticity [lines 62-70, 95-103, 431-440]. We also clarify the difference between signed contributions (which sum to estimate the difference in population growth rate between the target population m and the reference R) and effect magnitudes (which are the absolute values of contributions, scaled to sum to unity as % of total effect) [lines 205-210]. To illustrate the difference between elasticities and vital rate effects, we include a new Fig 4 – a ternary diagram that compares vital rate elasticities (heavily weighted toward child survival) vs. vital rate effects, which are much more diverse.

 We also clarified that we are talking about different approaches for explaining differences in population growth rates, within and between species [lines 73-76, 198-205]. We also take care to maintain tense agreement throughout, using the present tense for statistics (e.g. averaged vital rates) and prospective estimates (e.g. elasticities) and the past tense for contributions and effects, since they are retrospective decompositions of observed differences. Also, we added a very brief treatment of the difference between prospective and retrospective analyses and we now reference a more detailed treatment of this distinction (Horvitz et al. 2007) [lines 117-122]. 

The authors are later more explicit when referring to LTRE where they are defined as the “vital rate

contributions to observed differences in population growth rates” between two projection matrices (please note that vital rates are not individual measures as mentioned in l71 since there are population aggregates). In this sense, they are not “contribution to [a population] fitness” but how differences in entries of two matrices translate into difference in change of population reference growth rate. I strongly suggest the authors to define it more clearly. A way to do it is that sensitivity sij is the impact on λ of one unit of change in matrix entry aij. If we multiply sij by Δaij, it tells us how such a change would have modify the reference population growth λ.

Thanks for this comment. We modified the text to read “…decompose contributions of different vital rates to observed differences in population growth rates. Vital rate contributions (Cij) are estimated by multiplying vital rate sensitivities (sij), which reflects the fitness effect of a one-unit change in matrix element ai, by population-level differences (Δai) in vital rates (Δaij = aij(m) – aij(R); Cij = sij Δai), comparing each observed population (m) with a common reference (R).” [lines 88-93]. We agree that LTREs do not estimate contributions to population fitness unless you use a null matrix as the reference, in which case the contribution of matrix element aij is the product of the vital rate and the sensitivity (Cij = sij Δaij where Δaij = aij – 0, so Cij = sij aij ). In our LTRE, vital rate contributions (Cij) are estimated by multiplying vital rate sensitivities (sij) by population-level differences (Δai ) in vital rates (Δaij = aij(m) – aij(R); Cij = sij Δai), comparing each observed population (m) with a common reference (R), which contains the mean vital rates calculated across all hunter-gatherers.

Second, the authors then states l74-75 that “Differences between realized fitness contributions and the potential suggested by elasticities may indicate constraints on life history evolution” (also 405-406). This can be a fantastic idea and I can intuit what the authors have in mind. Yet it is not trivial to me, and it makes me wonder if this has been already theorized elsewhere. If it has, the author should clearly state it and explain why (I think not shying away equations). If it has not, I would strongly encourage the authors to develop - and if possible demonstrate - this idea. For instance, Cij, is a given amount of change between two matrix waited by sensitivities. Does this idea relates to the long lasting debate on the difference between using sensitivities and elasticities?

We appreciate the reviewer’s enthusiasm here. Saether and Bakke (2000) conjecture that LTRE contributions may be small despite large elasticities, due to stabilizing selection buffering important vital rates against temporal variation (Pfister 1998). Others have noted problems with using elasticities to predict vital rate effects and LTREs have been presented as the best tool for population comparison. We now include more background literature on this subject in the introduction, and include more implications of our findings in the discussion [lines 117-122]. Our Z metrics directly compare prospective vs. retrospective “importance” measures, but it is not trivial to derive inferences from low (<<1) vs. high (>>1) deviations from prospective estimates.

When we compare retrospective contributions to prospective elasticities, we ask what departures between these alternative measures of the relative “importance” of vital rates might tell us about selection or constraints on variation across populations. For instance, if the relative effects of child survival differences are smaller than their elasticities suggest, this could support the “buffering hypothesis” of Pfister (1998) that has been predicted (Saether and Bakke 2000) to reduce variation in child survival rates between populations as well as within populations over time. This would mean that child survival elasticities would be larger than the relative effects of child survival differences, whereas effects of adult survival and fertility differences would be larger than their elasticities. If contributions of fertility differences are larger than their elasticities, this may reflect fertility-survival tradeoffs, while larger contributions of adult survival may indicate reproductive tradeoffs that make up for high child mortality in uncertain or low-resource environments [lines 96-104].

As far as we understand, the debate between sensitivities and elasticities centers on the importance of structural zeros in the population matrix. For instance, human fertility is zero at age 5 in all populations and so the elasticity E16 will be zero - but the large sensitivities indicate the change in population growth that would occur if age 5 humans suddenly evolved non-zero fertility. This is an interesting possibility, and suggests that selection would be strong on age 5 fertility, but because there is zero variation in age 5 fertility rates, there is nothing for selection to work with (selection requires variation in heritable fitness-relevant traits).

Main comment 2

The authors used the ratio between contribution and elasticity to measure (if I understand well) these

possible constraints. But, I would strongly suggest the authors to check the resulting equation. First, l146, I think there is a mistake: eij is not equal to sij*(λ/aij) but to sij*(aij/λ) (I guess that this is a typo because elasticities look ok in fig 1).

Yes, thanks for catching this. We fixed the typo. [line 186]

But then Zij = Cij/Eij = (Δaij.sij)/(sij(aij/λ))=(Δaij/ aij)(1/λ).

Then Z is the ‘percentage’ of difference between the reference and the analysed matrices divided by the

growth rate. I am far on being clear on what does this mean and how this allows identifying constraint on a vital rate. I therefore strongly suggest the authors to explicit this metric and how/why it is used to solve their research question.

We try to explain this metric better and discuss the implications of deviations from 1:1 parity between prospective fitness elasticities and retrospective fitness contributions [lines 112-116, 236-242, 431-440, 579-582], including a new Fig 4 that illustrates the contrast between these metrics. Actually, Zij = Cij/Eij = (Δaij * sij) / (sij (aij / λ)) = (sij /sij) (Δaij / aij) (λ) = (Δaij/ aij)(λ), which reflects the effects of population-level differences, and so would be the proportional change in the vital rate scaled by the population growth rate (in numerator). We talk about Z as being the proportion of the potential vital rate effects that are realized by population-level differences (with both contributions Cij and elasticities Eij invoking sensitivities sij). Although this could be flipped around as you simplified it, to merely reflect the vital rate differences we are looking at how these vital rate differences are scaled by elasticities to drive LTRE contributions (and thus drive them to differ from elasticities that do not take vital rate differences into account). In essence Z reflects the variation across populations in a vital rate that are ignored by elasticities but are included in LTRE contributions predicting the fitness effects of observed vital rate differences. If Z if far below 1.0, then the “importance” estimates of elasticities fail to predict the vital rates actually driving differences we observe between populations. In this case, Z << 1 (C << E) might indicate constraints on vital rate variability due to stabilization selection-buffered traits and Z >>1 (C>>E) might indicate constraints on stabilizing (or directional) selection due to tradeoffs such as that between fertility and infant survival.

I am also not clear on whether Z should be sum(Cij)/sum(Eij) or rather sum(Cij/Eij), which can be substantially different. 

We use the sum(Cij)/sum(Eij) because it is consistent with how we report the proportion of all contributions or elasticities due to a given life cycle component [lines 205-210, 279-280].

Finally I don’ t understand the values for the Zs in table 1. For instance for Ache, Zc=Cc/Ec=7/42=0.16, not 95. Or, am I missing something?

This was a typo in the table, now fixed, where the column headers for Ca and Cc were switched. So for the Ache, Zc = Cc/Ec = 40/42 = 0.95, Za = Ca/Ea = 7/54 = 0.13. For consistency, we also added a Zs column for all survival rates where Zs = Cs/Es = 0.49. [line 289]

I would suggest to incorporate Table S5 into the main text.

Ok, we now incorporate the former Table S5 in the main text as Table 1 [lines 192-194].

Main comment 3

I am not sure that I understand prediction 1 and it may be there a conceptual mistake. Canalization is the fact that vital rates impacting the more fitness (here λ) should exhibit lower temporal variance than those under weaker selection. The authors rightfully quote [22] and [23] testing this by somehow correlating the estimation of the variation in time of matrix entries to the variation of λ (but variance is in time, not between populations, isn’t it?).

Yes. First, P1 depends primarily on the large elasticities of child survival more than due to low variance. The Z metrics (Z = C/E) compare contributions (C) to elasticities (E) to see how well prospective estimates (elasticities) predict important vital rates driving population-level differences (LTRE contributions). The only thing that hinges on the “space-for-time” substitution in this comparison is the prediction (Saether and Bakke 2000) that high-elasticity rates should vary less across populations because they are subject to canalizing/stabilizing selection in each population. A previous version of this manuscript looked in more detail at these predictions but we are not claiming here that spatial variation in vital rates is a good indicator of temporal variation within populations and we have limited time-series data to validate the “space-for-time” proxy (see Gurven and Davison 2019 for brief treatment of the temporal variation documented in a small subset of these populations). 

Note also that, if I am not mistaken, [23] performed elasticity analysis not LTRE (as suggested in sentence l84) such that the effect of variance on LTRE is also not that clear to me. Anyway, I cannot see how LTRE between populations (without temporal variance accounted for) can allow identifying life-history constraint and how the concept of canalization is involved into this. If I am mistaken, I strongly suggest the authors to make their point more clear.

Saether and Bakke 2000 conducted both prospective (elasticity) and retrospective (LTRE) analyses, and found that LTRE contributions decreased with vital rate sensitivity (suggesting demographic buffering via stabilizing selection sensu Pfister 1998).

We have added text to P1 to help clarify our prediction here. [lines 112-122, 224-228] Although stabilizing selection occurs over time, we are comparing populations and we are using between-population comparisons. Saether and Bakke (2000) predicted smaller LTRE contributions (between populations) and other researchers have used “space-for-time” substitution with varying success (Strier 2016), so P1 uses the data available to see how well prospective elasticities predict observed (retrospective) vital rate contributions. We also mention in the Discussion how child survival actually varies a lot over time despite presumably strong stabilizing selection, but our main finding for P1 is that child survival differences have a smaller effect across populations than predicted by their high elasticities, whereas fertility differences have a larger effect. 

Main comment 4

I find that P1 (l81-82) is not well formulated. If I am not mistaken, it is a property of elasticity to be strictly declining with age in an age-structured model, infant and children survival elasticity always being constant and the largest. Metric have to be twisted and parameters very different that those of mammals to find alternative pattern (Baudish, 2005, PNAS). It is between species that relative magnitude of elasticities can be compared and I would strongly suggest to cite Heppell et al., 2000, Ecology for a comparison in mammals across the slow-fast continuum. Also why not refering to and using a classical Silvertown triangle to represented this (Silvertown, J., et al. 1993. Jouranl of Ecology 81:465–476)?

We now reference Heppel et al. 2000 when talking about age-patterns of elasticities differing within and between species [lines 131-133]. Whereas both the sensitivity and elasticity of survival decline with age due to declining expected future reproduction, elasticities are scaled by mean vital rates. This means that fertility elasticities are zero until AFB and then rise to a peak at some adult age, then decline due to mortality and and diminishing future fertility (Fig 1B).

We compare fitness contributions between populations and between species to see whether newborn survival (p0) makes the largest contribution because elasticity to recruitment is the largest elasticity. However, under strong buffering selection resulting in canalization, variation in newborn survival would be reduced, and thus fitness contributions from newborn survival should be small. Instead, we find large contributions of newborn survival, meaning that this vital rate is an important driver of population growth differences. This suggests that infant survival may not be buffered against variation, since Saether and Bakke (2000) predict smaller contributions from vital rates with high elasticities (Pfister 1998).

I am not sure what the authors want to test with prediction 2 which is the obvious fact that both increasing survival and fertility should increase population growth rate. Evidencing trade-off between fertility and survival?

Yes, exactly. Ordinarily you’d expect both higher survivorship and fertility to lead to higher growth rates – certainly this is the trivial case within a population, but does higher survivorship and fertility meaningfully predict higher growth rates across populations? They might not if survivorship benefits are among post-reproductive adults, or if there are trade-offs in vital rates. Such trade-offs have been documented in humans and non-human primates [24-28]. Our finding that population growth is decoupled from fertility in chimpanzees is consistent with a strong effect of mortality limiting potentially high fertility (if mortality were lower, high fertility would increase population growth more); that population growth is decoupled from survival in humans is consistent with long post-reproductive lifespans (contributing to high e0) during which direct fitness contributions are zero (surviving beyond ALB does not provide direct fitness).

As you suggest, we now include a ternary diagram (Fig 4), similar to those used by Silvertown 1993, as a more reader-friendly way of comparing elasticities for child survival, adult survival and fertility across populations. Fig 4A shows the ordination for elasticities (Ec, Ea, Ef), showing the strong selection on child survival and Fig 4B shows the ordination for vital rate effects (Cc, Ca, Cf), which are much more diverse and illustrate the difference between the potential for vital rate effects estimated by prospective elasticities and the observed vital rate effects, which scale elasticities by population-level differences.

Main comment 5

I would suggest the authors to discuss limitations of elasticities analyses in general and apply to humans in particular. (1) First elasticities are only one hand of the evolutionary GxE equation (Lande 1982;

Charlesworth 1990; Steppan et al. 2002). Evolution also need genetic variance and this could be

acknowledged. (2) The authors are comparing Leslie matrix, but any sub structuring (as individual

heterogeneity) or hidden trade-offs may change the results. (3) It the most important, it has been shown that intergenerational transfers between age-class or parental investment can strongly impacts elasticities on survival and fertility in humans (Lee 2003, PNAS, Pavard et al. 2007, Evolution, Pavard & Branger 2012 Theo Pop Biol). For instance, magnitude of elasticities on adult survival may be strongly underestimated when maternal or grand-maternal care is not implemented. Elasticities on fertilities by age can also exhibit very different patterns. Because such intergenerational transfers have been proposed as a very important drivers of the evolution of human life-history, the authors should at least discuss it. (4) As the authors wonderfully argued in a recent article, only periodic catastrophes in humans can explain the human forager paradox. It also means that all elasticity analysis in constant and infinite environment is somehow incomplete and elasticity should be considered into a stochastic model.

 We thank the reviewer for these insightful comments. 

(1) We now mention the need for genetic/phenotypic variability for natural selection to act on (and relate to canalization resulting small differences between populations, sensu Saether and Bakke 2000). [lines 514-515]

(2) Our asymptotic analyses look at the effects of changes in mean vital rates and do not address temporal variability in vital rates or the effects of individual variation. Although methods exist to decompose stochastic contributions (Davison et al. 2014), and we do reference temporal variation [lines 228-234] we do not have sufficient time-series date to conduct such analyses [lines 528-531].

(3) We mention the importance of intergenerational transfers driving indirect fitness contributions, and how intergenerational transfers can greatly alter the force of selection reflected in vital rate elasticities [lines 517-525]. However, estimating their fitness effects is not tractable with current methods, though we now cite Pavard et al. 2007 and Pavard & Branger 2012 for examples showing how transfers can impact elasticities in humans [lines 521-523]. We are excited to report that another paper in progress will present a new framework for estimating these indirect fitness contributions made via production or information transfers.

(4) Again, the existing subsistence population data limit our ability to conduct stochastic analyses but we acknowledge the importance of both demographic and environmental stochasticity that is missed in our analysis of averaged rates [lines 528-531]. We also miss individual heterogeneity that would be reflected in the underlying individual data but are obscured by our focus on average population statistics.

Minor comments & Détails

l418-420 – Isn’t there a contradiction is stating that juvenile survival is under canalization effect and stating later on that it varies more in time than adult survival?

Yes – if juvenile survival is canalized, then we shouldn’t expect it to vary much over time. In that case, our demonstration of substantial juvenile survival differences suggests that canalization is weak, or subject to environmental plasticity. We now clarify this point better in the paper to reduce any potential confusion [lines 224-235]. Juvenile survival is predicted to be canalized based on large elasticities but in fact it is highly variable (both over time as cited, and across populations as we show). This suggests limits on the ability of natural selection to buffer this rate (and we discuss why in terms of bet-hedging costs of reproduction).

Figure 1 – I am not sure how the SEM of elasticity is calculated. Is this trivial?

SEM = std(x)/sqrt(N), with std(x) taken across populations in a given set (e.g. hunter gatherers) and N being the number of populations in that set (e.g. 5 hunter-gatherers). 95% confidence intervals are represented by mean ± (2 SEM).

I am not sure what figure 2 really brings to the article since it is complicated and barely discussed.

Fig 2 summarizes the population-level net effects of survival vs. fertility. In the revised version, we explain its components better and discuss it more thoroughly in the paper [lines 336-343]. 

Problem in legend of figure 2 – Non-Forager are filled-circled as HG not filled square. Indicate that isolines are population growth rates. Remove the title.

Thanks for catching these. Typo corrected. Added reference to isoclines, and removed title. [lines 345-357]

L290 – I guess this is fig3.B instead of 2B,3?

It meant figure 2B and figure 3. Clarified in text. [lines 360, 364]

l 32 – I am not sure that reference [2] did anything in calculating the divergence time between humans and chimpanzees. Please check carefully this reference. I think it should instead be referred to l 33-34.

Thanks for your keen observation. We now more appropriately reference Hobolth et al. 2007 and Langergraber et al. 2012 there. [line 35]

l 35, “human fertility is similar to chimpanzees” and further. Please be more specific. Do you mean the shape of the age-specific fertilities? If yes, both the distribution and the TFR? Is the whole shape the distribution identical? The authors refer to [6] who focus mainly on reproductive senescence and show that if the timing of reproductive senescence is similar rate of reproductive senescence is not the same as well as how it correlates with decline in survival. I suggest to be more precise.

We added some remarks to better clarify similarities and differences in human and chimpanzee fertility [lines 37-40]. We cite [6] because of its supplement that includes ASFRs for multiple wild chimpanzee groups. In general, ASFR is similar during prime fertility ages but comparing them, on average, does reveal some key differences. Chimpanzees start reproducing at earlier ages and some continue to reproduce at later ages than humans (although most chimpanzees do not live until these later ages). In addition to fairly similar age profiles, which have been noted before, we find no significant difference in TFR, pointing out that completed chimpanzee fertility would be similar to that of human hunter-gatherers if they had human-like mortality schedules. 

L37-38 – “However, there is great variation among human and chimpanzee life histories”. Here again I suggest to be more specific. The authors quote [8]. Although a valid reference, it can be completed by more recent article (as the [2]). Furthermore, I am not a native speaker but is “difference” would be better than “variation”?

We add the reference [2], now [9] [line 42]. To clarify, we are looking here at the degree of variation in vital rates (shown in the SEM envelopes), which shows higher variation across populations among humans vs. chimpanzees. This variation is due to differences in many vital rates that drives differences in population growth rates. Whereas our LTRE contributions examine the effects of life history differences, here we are characterizing the degree of variation among our study populations.

L39 and many time after – Please change “within species” by between population. In ecology, within species study refers more to the study of variance between individuals than between population as it is investigate here.

We are not investigating individual-level variation but rather variation at the population level, within- vs. between- species. The variation across human populations is used to estimate the variation within humans, whereas species differences are investigated by comparing mean life histories of humans vs. chimpanzees. To avoid confusion, we clarify this distinction the first time it is introduced. [line 45-47]

L 40 - “We interpret population life histories in terms of the slow-fast life history continuum [9]” – Why? Also, human a complete outlier on this continuum so that I wonder if this is relevant.

Others have shown that primates are “slow” compared to other mammals, that chimpanzees are “slow” compared to other primates and that humans are “slow” compared to chimpanzees, so humans could well be seen as an outlier. As others have argued before us, even among humans there is a fast-slow continuum. At the same time that humans can be thought of as having “slow” life histories in comparative light, humans may also be considered outliers in the sense that they combine elements of slow life histories (longevity, delayed maturity) and elements of fast life histories (high fertility, short IBI). We cite other references for slow/fast LH in primates/chimpanzees/humans. [lines 42-45, 124-130, 517-521]

L41 – To my knowledge Stearns’ book (but I don’t have it at hand here to check) is about trade-offs in

general not about their importance for human life-history evolution.

True, we are extending the idea of life history tradeoffs to humans. Others (e.g. Gillespie et al. 2008; Lawson et al. 2012) show costs of reproduction in humans and so we cite that work here instead [lines 124-130, 510-513]

In [6] the authors use extensive data in chimpanzees. Yet, this represent only about 600-1000 individuals (the equivalent of a small human village) spread in small groups over nearly a continent. How this could affect the authors’ results?

The small sample sizes may not accurately represent the diversity of chimpanzee life histories (or the diversity of individual life courses among chimpanzees), but they are the best data available. The ASFRs are fairly similar across groups. Thus, in [6] the authors provide ASFRs in two ways: averages across populations, and combined as if all from the same population (i.e. equivalent to weighted by sample size). The differences in ASFRs across these two conditions are minimal. 

L55 – I am not found of the concept of population fitness underlying in this sentence.

Does this mean the reviewer is skeptical about the utility of elasticities for estimating the force of selection? This has a long tradition, but there is also a debate of the usefulness of sensitivities vs. elasticities

Elasticities and sensitivities are both useful in that they tell us about the potential for fitness effects if vital rates change (a measure of “bang-for-your-buck” in terms of fitness changes when vital rates differ). Elasticities have the additional utility of being proportional (a % increase in population growth due to a % increase in a vital rate); by scaling differences by their mean values elasticities allow us to compare the relative effects of changes in fertility vs. survival on the same y-axis. LTRE contributions, on the other hand, scale fitness effects to the differences between populations, so small differences in a high-elasticity rate can make larger fitness contributions than large differences in a low-elasticity rate. LTRE contributions, therefore, show us what actually explains differences in population growth rates based on the observed vital rate differences and their associated elasticities. Comparing realized (retrospective) LTRE contributions to potential suggested by (prospective) elasticities may tell us about potential constraints on selection when vital rate variation is limited (due to inherently low genetic variation or stabilizing selection being stronger than directional selection).

L71 – Indicates the pages in [15]. Note that you could have also quoted Hal Caswell, 1989, Analysis of life table response experiments I. Decomposition of effects on population growth rate, Ecological Modelling, Volume 46, Issues 3–4.

We now cite Caswell 1989 [line 121]

L153 – this should be sij instead of sj isn’t it?

Typo fixed [line 186]

L179 – I am not sure to understand why the fact that Cij and Eij sum to unity allow to calculate the ratio.

Because Cij and Eij each sum to unity, they represent the proportion of all observed effects and the proportion of potential effects, respectively. Therefore, we can sum the proportion of effects due to child survival and the proportion of elasticities to child survival and see if a larger or smaller proportion of retrospective effects was observed, relative to the potential suggested by prospective elasticities. This contrast is now illustrated by a new Fig 4.

We acknowledge that ratios of ratios are sometimes confusing or inappropriate, but here the ratio tells us whether proportional LTRE contributions due to X are greater than the proportional elasticity effects due to X. Because we already examine the net LTRE contributions (Fig. 3A), we look at effect magnitudes, which do not obscure opposing contributions (C < 0 vs C > 0), and we scale each effect magnitudes relative to the sum of all effect magnitudes because this makes them directly comparable to elasticities (which are likewise scaled to sum to unity across all the potential effects). [lines 205-210].

Figure 3A – I find the figure very complicated to figure out. Are they the mean summed C values between populations? Then why and how is separated positive and negative C values? Or “composite” refers to the mean trajectories for HG, F, WC. But then, again, how does it lead to both (+) and (-) for a same trait (i.e., Infant survival). I am very sorry if I miss this information.

Fig 3A shows net results for contributions (summing to the difference between a population growth rate and that of the mean hunter-gatherer reference), vs. 3B showing the relative effect magnitudes (summing to unity to be comparable with elasticities – see the new Fig 4). These effect magnitudes are what we use to test for differences in means and to calculate the Z ratios (for P1).

In Fig 3A, the positive values are summed above the origin and the negative values summed below, illustrating opposing contributions that, on net, yield the difference in population growth rate (inset white bars). For instance, Infant survival is from age 0 to age 2 so there are two age-contributions (a21 for survival 0 to 1, a32 for survival 1 to 2) that may be opposite in sign. We make a note in the figure caption explaining why contributions for a given life cycle component (e.g. infant survival) can have opposing contributions above and below the origin (C = 0). [lines 370-373]

The “composite” populations in Fig. 3A are the synthetic populations with vital rates equal to the average over a given group (e.g. the LTRE reference life history with vital rates averaged across hunter-gatherers, HG). Fig. 3B legend now clarifies that effect magnitudes averaged across groups of individual populations (e.g. the 5 hunter-gatherer populations), rather than results for the mean life histories (e.g. the mean hunter-gatherer reference) [lines 377-380].

Fig 3A shows vital rate contributions that are summed across life history components (infant, child and adult survival; early, prime and late fertility), but the positive and negative components are summed and plotted separately (positive contributions above zero and negative contributions upside-down below zero). If individual vital rates within a single life history component may have different signs they will appear both above and below the zero line (e.g. infant survival is from age zero to age 2 and includes both newborn survival age 0 to 1 and survival age 1 to 2, so it may have both positive and negative contributions). In Fig 3B these opposing contributions are weighted equally in the combined magnitude so they sum to unity (100% of all effect magnitudes).

---

## [Editor Report · Decision Letter 1]

5 Jan 2021

PONE-D-20-26560R1

Human uniqueness? Life history diversity among small-scale societies and chimpanzees

PLOS ONE

Dear Dr. Davison,

Thank you for submitting your manuscript to PLOS ONE. After careful consideration, we feel that it has merit but does not fully meet PLOS ONE’s publication criteria as it currently stands. Therefore, we invite you to submit a revised version of the manuscript that addresses the points raised during the review process.

Reading the revised manuscript and the replies to reviewers’ comments, I feel the concerns raised by the two reviewers were addressed satisfactorily. However, I would like to suggest some further improvements (I will not insist neither of the suggestions).

Reviewer 1 included some questions. Those were answered in the replies but were not reflected in the manuscript. The questions were treated as minor comments, but the readers might have the same questions. I am wondering if a couple of sentences should be added in the manuscript to clarify (e.g. by citing the papers that were used in the replies).

Tables 2 & 3 are difficult to digest. It is better to convert them into figures and move the tables to appendix.

We look forward to receiving your revised manuscript.

Kind regards,

Masami Fujiwara, PhD

Academic Editor

PLOS ONE

---

## [Author Response · Author response to Decision Letter 1]

22 Jan 2021

Response to Editor’s comments:

Dear Dr. Davison,

Thank you for submitting your manuscript to PLOS ONE. After careful consideration, we feel that it has merit but does not fully meet PLOS ONE’s publication criteria as it currently stands. Therefore, we invite you to submit a revised version of the manuscript that addresses the points raised during the review process.

Reading the revised manuscript and the replies to reviewers’ comments, I feel the concerns raised by the two reviewers were addressed satisfactorily. However, I would like to suggest some further improvements (I will not insist neither of the suggestions).

Reviewer 1 included some questions. Those were answered in the replies but were not reflected in the manuscript. The questions were treated as minor comments, but the readers might have the same questions. I am wondering if a couple of sentences should be added in the manuscript to clarify (e.g. by citing the papers that were used in the replies).

We have now clarified a number of passages to better address the concerns of reviewers that were answered in the Response but not directly addressed in the text.

R1 Lines 213-227: We now clarify why ALB may be lower among managed chimpanzees (Lines 154-158).

R1 Lines 213-227: We now cite Van Groenendael et al. (1994) as a reference for why E0 = Ef (Lines 527-530).

R2 Main Comment 4: We have clarified the importance of elasticities and fitness effects summing to unity and following Silvertown et al. (1993, now referenced) we have included a new ternary diagram contrasting these two metrics (lines 210-213) 

R2 Main Comment 5: We have added additional references addressing the possibility that negative genetic correlations could limit vital rate contributions (Lines 525-527).

Tables 2 & 3 are difficult to digest. It is better to convert them into figures and move the tables to appendix.

Table 2: We have now put most of the results shown in Table 2 into a new Fig 2 (Lines 276-287). The top panel (A) shows mortality statistics (lα, lM, lω, e0) and the bottom panel (B) shows fertility statistics (AFB, ALB, IBI, TFR, and rough estimates of ages at parity 0-10 assuming equal birth spacing IBIs between AFB and ALB). The remaining Table 2 only contains the Z metrics comparing observed fitness effects to the potential described by elasticities (Lines 446-454).

Table 3: We have simplified Table 3 by reducing the number of decimal places reported and converting the significance metrics from columns containing p-values to asterisks (Lines 465-472).

We look forward to receiving your revised manuscript.

Kind regards,

Masami Fujiwara, PhD

Academic Editor

PLOS ONE

 

RESPONSE TO REVIEWERS

[R1] Reviewer Comments for PONE-D-20-26560

Human uniqueness illustrated by life history diversity among small-scale societies and chimpanzees

Authors of this manuscript showed that delayed maturity and adult mortal-

ity is the main difference to separate humans from chimpanzees which shares

common ancestor. They employed Life Table Response Experiments to quantify

vital rate contributions to population growth rate. Their results and discussion

are is interesting. Their approach is justified. However, this manuscript con-

tains numerous inconsistencies in its data and statements. It requires careful

attention on details to justify its results and conclusions.

We thank you for the positive feedback. We have revised the paper considerably to account for all of the reviewers’ careful comments. Our responses are in bold.

Major inconsistencies and questions:

• Lines 213 - 227, Table 1: In the manuscript, line 180 states that Za =

Ca=Ea and lines 148 and 150 shows that Ec + Ea + Ef = 1. However,

numbers in the table are inconsistent to the statement in text. Za 6=

Ca=Ea and Ec + Ea + Ef 6= 1

We apologize for the typo in which the column headers for Cc and Ca were switched. [Line 289]

We have now made sure all notation is consistent throughout the paper: 

(Ec + Ea) + Ef = Es + Ef = 1 in every row, and in every row Z = C/E (e.g. Zf = Cf/Ef).

• Lines 213 - 227, Table 1: Why E0 = Ef ?

This is a fact that is demonstrated by loop elasticities, which show that the sum of elasticities coming into a life history stage must be equal to the sum of elasticities across all outgoing transitions. Here, E0 = E(a21) = sum(Efert) because the elasticity to recruitment (the only pathway out of the newborn “stage”) is equal to the sum of elasticities to fertility at different ages (Van Groenendael et al. 1994).

Van Groenendael, Jan, Hans De Kroon, Susan Kalisz, and Shripad Tuljapurkar. "Loop analysis: evaluating life history pathways in population projection matrices." Ecology 75, no. 8 (1994): 2410-2415.

• Lines 213 - 227, Table 1: Why do managed and captive population have

smaller ALB compare to Wild population for chimpanzees?

For the captive breeding program at Taronga Zoo, ALB estimates come from only 7 females, including one that died 3.5 y after the last birth, one on contraception, and one that was transferred (Littleton 2005), so this is likely an underestimate of captive ALB. In the Gambia population, they do not use contraceptives and AFBs are similar to wild chimpanzees, but there may be factors stemming from prior captivity that limit their reproductive lifespans (Marsden et al. 2006). These are some of the reasons that only wild chimpanzees are considered in the ALB comparisons (S2 table).

• Fig 3A: Why are there two yellow and red sections in the bar of WC- and

WC? Why are there two red and brown sections in the bar of WC+? Why

does Infant survival (the purple section) have both positive and negative

contributions in WC- and WC?

The bars above the y-axis origin (C = 0) show positive contributions and those below the origin show negative contributions. They sum together to the net contributions (shown as white bars), which sum to the total difference in population growth rate between the target (m) and Reference (R). This is clarified in the figure caption. [lines 370-373].

We consider infancy as from birth (age 0) to 2 yrs. Thus, there are two age-contributions of infant survival (p0 = a21, p1 = a32) that may be of opposing sign. For instance, with WC, newborn survival p0 (a21) makes positive contributions because it is higher than that of hunter-gatherers (R), but older infant survival p1 (a32) makes a negative contribution because age 1-2 survival estimates for the WC average and WC- (declining chimpanzee average) are lower than those of the hunter-gatherer average (R).

• Lines 337-338: \\E0 > Ec > Ea > Ef " is not consistent with numbers in Table 1.

We regret the confusion caused by these inequalities and we have simplified our explanation of P1, removing the inequality and explaining the prediction in clearer English. [line 416]

E0 (the elasticity of newborn survival p0, which could also be written as E21, the elasticity of matrix element a21) is the largest elasticity for a single transition (here, individual matrix elements). Ex+1,x<α is larger than Ex+1,x≥α at all ages and Ex+1,x > E1x except for the last couple years of reproductive life. However, the values in Table 1 are the sums across all ages (e.g. Ec = Σx<α Ex+1,x, Ea = Σx≥α Ex+1,x), so Ea may be larger than Ec because it sums over a larger range of ages (all ages after α, the minimum AFB). As stated above, E0 = Ef = Σx E1x , requiring that the elasticities to fertility at single ages are each lower than E0.

• Line 345: \\Cc > Ca" is not consistent with numbers in Table 1.

We have fixed a typo where the column headers for Cc and Ca were switched in Table 1. [lines 289, 423]

• Line 350: \\Cs _ Cf " is not consistent with numbers in Table 1.

This “≈” indicates there is no significant difference between Cs and Cf (p > 0.1), not that these values are equal. [line 429]

• Lines 357-359: \\Zc _ Za", \\Zc < Za" and \\Zc < Za" are not consistent with numbers in Table 1.

The inequalities were reversed and are now correct and in agreement with Table 1: [lines 437-442]

Zc ≈ Za (HG, p > 0.1), Zc > Za (WC, p = 0.016), Zc > Za (NF, p = 0.095), Zc << Zf (p ≤ 0.008).

We now clarify that, by saying that adult survival effects are more under-estimated than child survival effects, this means that Za if farther below 1:1 than Zc (so Zc > Za). 

Minor points:

• Line 127: please spell out \\NLIN".

Done [line 165]

• Lines 131 to 133: Is it a duplicated statement?

This statement is not duplicated elsewhere in the paper.

• Line 146: \\(eij = (@_=@aij) = sij(_=aij)" should be \\(eij = (@_=@aij) =

sij(aij=_))".

Typo fixed [line 186]. We also capitalize all elasticities (e.g. Eij) to avoid confusion with the lower-case e0 for life expectancy [lines 185-191].

• Line 153: \\ sumi;j sj_aij" should be \\ sum_i;j (sij_aij).

Typo fixed [line 197]

• Line 184: \\maxi;j(eij)" should be `max(eij)".

We now clarify that E0 (distinguished from life expectancy e0 by capitalization) is the highest elasticity (E0 = max(Eij)) – elasticity to age 0 survival in the matrix element a21. [lines 245-246]

• Table 1 row 1: Please add description of l_, lM, l!, TFR, AFB, MAC,

ALB and IBI in the caption.

Done [lines 274-277]. We also include a new Table 1 with variable definitions and source equations [lines 192-194, moved from Supporting Information].

• Table 1 row 1: Please switch column Ca and Cc for display consistency.

Done [line 289]

• Line 278, Fig 2: Cannot _nd the \\non-foragers by _lled squares" in Fig 2.

Not sure why. They were in the original file and we have made sure they are in the current file. [lines 347-349]

• Line 279, Fig 2: Cannot _nd the \\(labeled NF)" in Fig 2.

Again, not sure why. It is the unfilled square with a dot in it, located at (-0.001157,0.01994). 

• Line 349: \\S4 Table" should be "S2 Table".

Fixed to say S3 Table (Differences within populations) [line 425]

• Line 352: \\S4 Table" should be "S2 Table".

Fixed to say S3 Table (Differences within populations) [line 430]

• Line 360: \\S4 Table" should be "S2 Table".

Fixed to say S3 Table (Differences within populations) [line 442]

• It will be helpful to add a data table of computed age specific mortality

and fertility of each population (which are the data used to plot Fig 1)

into online Supporting Information (S2, S3 Tables).

Added [lines 882-887]

 

[R2] Review of Human uniqueness illustrated by life history diversity among small-scale societies and chimpanzees by Raziel Davison and Michael Gurven.

In this article, the authors compare age-trajectories of survival and fertility of many hunter-gatherer and

forager human populations and chimpanzees. They then perform LTRE and spectral analysis to investigate potential evolutionary changes between these trajectories. This is a very interesting manuscript, revisiting with originality and up-to-date data a classic question in human evolutionary biology: the evolution of the human and chimpanzee life-cycle since divergence. The manuscript incorporates a wonderful comparison of age-trajectories of the different not-industralized populations with several chimpanzee populations, again using the best data to date. In this respect I think that this article has great potential. I am more skeptical about the evolutionary interpretation of the LTRE and elasticity analysis. I think that there are several conceptual and technical issues which need to be addressed prior to publication.

We appreciate the positive comments. Below we address the reviewer’s concerns about conceptual and technical issues.

Main comments 1

The authors mainly use two metrics in the analyses: elasticities and what they called fitness contributions. All over the manuscript, I found unclear the definition of fitness contribution, and what the two measures together brings to the analyse. 

First, I think there is a problem of definition throughout the manuscript. This starts L 52-55 where I found the sentence “These fitness contributions illustrate how life history event schedules drive differences within and between species, while elasticities reflect the force of selection and highlight the potential for fitness contributions if vital rates vary across populations [19]” not very clear (illustrate? drive difference of what on what?). The authors quote [19] which focuses (to my knowledge) exclusively on elasticity and tells anything (as far as I remember it) on contrasting elasticity and ‘fitness contribution’. 

We now more carefully define “fitness contribution” [lines 88-95, 195-200], and clarify the distinction between this measure and fitness elasticity [lines 62-70, 95-103, 431-442]. We also clarify the difference between signed contributions (which sum to estimate the difference in population growth rate between the target population m and the reference R) and effect magnitudes (which are the absolute values of contributions, scaled to sum to unity as % of total effect) [lines 205-210]. To illustrate the difference between elasticities and vital rate effects, we include a new Fig 4 – a ternary diagram that compares vital rate elasticities (heavily weighted toward child survival) vs. vital rate effects, which are much more diverse.

 We also clarified that we are talking about different approaches for explaining differences in population growth rates, within and between species [lines 73-76, 198-205]. We also take care to maintain tense agreement throughout, using the present tense for statistics (e.g. averaged vital rates) and prospective estimates (e.g. elasticities) and the past tense for contributions and effects, since they are retrospective decompositions of observed differences. Also, we added a very brief treatment of the difference between prospective and retrospective analyses and we now reference a more detailed treatment of this distinction (Horvitz et al. 2007) [lines 117-122]. 

The authors are later more explicit when referring to LTRE where they are defined as the “vital rate

contributions to observed differences in population growth rates” between two projection matrices (please note that vital rates are not individual measures as mentioned in l71 since there are population aggregates). In this sense, they are not “contribution to [a population] fitness” but how differences in entries of two matrices translate into difference in change of population reference growth rate. I strongly suggest the authors to define it more clearly. A way to do it is that sensitivity sij is the impact on λ of one unit of change in matrix entry aij. If we multiply sij by Δaij, it tells us how such a change would have modify the reference population growth λ.

Thanks for this comment. We modified the text to read “…decompose contributions of different vital rates to observed differences in population growth rates. Vital rate contributions (Cij) are estimated by multiplying vital rate sensitivities (sij), which reflects the fitness effect of a one-unit change in matrix element ai, by population-level differences (Δai) in vital rates (Δaij = aij(m) – aij(R); Cij = sij Δai), comparing each observed population (m) with a common reference (R).” [lines 88-93]. We agree that LTREs do not estimate contributions to population fitness unless you use a null matrix as the reference, in which case the contribution of matrix element aij is the product of the vital rate and the sensitivity (Cij = sij Δaij where Δaij = aij – 0, so Cij = sij aij ). In our LTRE, vital rate contributions (Cij) are estimated by multiplying vital rate sensitivities (sij) by population-level differences (Δai ) in vital rates (Δaij = aij(m) – aij(R); Cij = sij Δai), comparing each observed population (m) with a common reference (R), which contains the mean vital rates calculated across all hunter-gatherers.

Second, the authors then states l74-75 that “Differences between realized fitness contributions and the potential suggested by elasticities may indicate constraints on life history evolution” (also 405-406). This can be a fantastic idea and I can intuit what the authors have in mind. Yet it is not trivial to me, and it makes me wonder if this has been already theorized elsewhere. If it has, the author should clearly state it and explain why (I think not shying away equations). If it has not, I would strongly encourage the authors to develop - and if possible demonstrate - this idea. For instance, Cij, is a given amount of change between two matrix waited by sensitivities. Does this idea relates to the long lasting debate on the difference between using sensitivities and elasticities?

We appreciate the reviewer’s enthusiasm here. Saether and Bakke (2000) conjecture that LTRE contributions may be small despite large elasticities, due to stabilizing selection buffering important vital rates against temporal variation (Pfister 1998). Others have noted problems with using elasticities to predict vital rate effects and LTREs have been presented as the best tool for population comparison. We now include more background literature on this subject in the introduction, and include more implications of our findings in the discussion [lines 117-122]. Our Z metrics directly compare prospective vs. retrospective “importance” measures, but it is not trivial to derive inferences from low (<<1) vs. high (>>1) deviations from prospective estimates.

When we compare retrospective contributions to prospective elasticities, we ask what departures between these alternative measures of the relative “importance” of vital rates might tell us about selection or constraints on variation across populations. For instance, if the relative effects of child survival differences are smaller than their elasticities suggest, this could support the “buffering hypothesis” of Pfister (1998) that has been predicted (Saether and Bakke 2000) to reduce variation in child survival rates between populations as well as within populations over time. This would mean that child survival elasticities would be larger than the relative effects of child survival differences, whereas effects of adult survival and fertility differences would be larger than their elasticities. If contributions of fertility differences are larger than their elasticities, this may reflect fertility-survival tradeoffs, while larger contributions of adult survival may indicate reproductive tradeoffs that make up for high child mortality in uncertain or low-resource environments [lines 96-104].

As far as we understand, the debate between sensitivities and elasticities centers on the importance of structural zeros in the population matrix. For instance, human fertility is zero at age 5 in all populations and so the elasticity E16 will be zero - but the large sensitivities indicate the change in population growth that would occur if age 5 humans suddenly evolved non-zero fertility. This is an interesting possibility, and suggests that selection would be strong on age 5 fertility, but because there is zero variation in age 5 fertility rates, there is nothing for selection to work with (selection requires variation in heritable fitness-relevant traits).

Main comment 2

The authors used the ratio between contribution and elasticity to measure (if I understand well) these

possible constraints. But, I would strongly suggest the authors to check the resulting equation. First, l146, I think there is a mistake: eij is not equal to sij*(λ/aij) but to sij*(aij/λ) (I guess that this is a typo because elasticities look ok in fig 1).

Yes, thanks for catching this. We fixed the typo. [line 186]

But then Zij = Cij/Eij = (Δaij.sij)/(sij(aij/λ))=(Δaij/ aij)(1/λ).

Then Z is the ‘percentage’ of difference between the reference and the analysed matrices divided by the

growth rate. I am far on being clear on what does this mean and how this allows identifying constraint on a vital rate. I therefore strongly suggest the authors to explicit this metric and how/why it is used to solve their research question.

We try to explain this metric better and discuss the implications of deviations from 1:1 parity between prospective fitness elasticities and retrospective fitness contributions [lines 112-116, 236-242, 431-442, 581-583], including a new Fig 4 that illustrates the contrast between these metrics. Actually, Zij = Cij/Eij = (Δaij * sij) / (sij (aij / λ)) = (sij /sij) (Δaij / aij) (λ) = (Δaij/ aij)(λ), which reflects the effects of population-level differences, and so would be the proportional change in the vital rate scaled by the population growth rate (in numerator). We talk about Z as being the proportion of the potential vital rate effects that are realized by population-level differences (with both contributions Cij and elasticities Eij invoking sensitivities sij). Although this could be flipped around as you simplified it, to merely reflect the vital rate differences we are looking at how these vital rate differences are scaled by elasticities to drive LTRE contributions (and thus drive them to differ from elasticities that do not take vital rate differences into account). In essence Z reflects the variation across populations in a vital rate that are ignored by elasticities but are included in LTRE contributions predicting the fitness effects of observed vital rate differences. If Z if far below 1.0, then the “importance” estimates of elasticities fail to predict the vital rates actually driving differences we observe between populations. In this case, Z << 1 (C << E) might indicate constraints on vital rate variability due to stabilization selection-buffered traits and Z >>1 (C>>E) might indicate constraints on stabilizing (or directional) selection due to tradeoffs such as that between fertility and infant survival.

I am also not clear on whether Z should be sum(Cij)/sum(Eij) or rather sum(Cij/Eij), which can be substantially different. 

We use the sum(Cij)/sum(Eij) because it is consistent with how we report the proportion of all contributions or elasticities due to a given life cycle component [lines 205-210, 279-280].

Finally I don’ t understand the values for the Zs in table 1. For instance for Ache, Zc=Cc/Ec=7/42=0.16, not 95. Or, am I missing something?

This was a typo in the table, now fixed, where the column headers for Ca and Cc were switched. So for the Ache, Zc = Cc/Ec = 40/42 = 0.95, Za = Ca/Ea = 7/54 = 0.13. For consistency, we also added a Zs column for all survival rates where Zs = Cs/Es = 0.49. [line 289]

I would suggest to incorporate Table S5 into the main text.

Ok, we now incorporate the former Table S5 in the main text as Table 1 [lines 192-194].

Main comment 3

I am not sure that I understand prediction 1 and it may be there a conceptual mistake. Canalization is the fact that vital rates impacting the more fitness (here λ) should exhibit lower temporal variance than those under weaker selection. The authors rightfully quote [22] and [23] testing this by somehow correlating the estimation of the variation in time of matrix entries to the variation of λ (but variance is in time, not between populations, isn’t it?).

Yes. First, P1 depends primarily on the large elasticities of child survival more than due to low variance. The Z metrics (Z = C/E) compare contributions (C) to elasticities (E) to see how well prospective estimates (elasticities) predict important vital rates driving population-level differences (LTRE contributions). The only thing that hinges on the “space-for-time” substitution in this comparison is the prediction (Saether and Bakke 2000) that high-elasticity rates should vary less across populations because they are subject to canalizing/stabilizing selection in each population. A previous version of this manuscript looked in more detail at these predictions but we are not claiming here that spatial variation in vital rates is a good indicator of temporal variation within populations and we have limited time-series data to validate the “space-for-time” proxy (see Gurven and Davison 2019 for brief treatment of the temporal variation documented in a small subset of these populations). 

Note also that, if I am not mistaken, [23] performed elasticity analysis not LTRE (as suggested in sentence l84) such that the effect of variance on LTRE is also not that clear to me. Anyway, I cannot see how LTRE between populations (without temporal variance accounted for) can allow identifying life-history constraint and how the concept of canalization is involved into this. If I am mistaken, I strongly suggest the authors to make their point more clear.

Saether and Bakke 2000 conducted both prospective (elasticity) and retrospective (LTRE) analyses, and found that LTRE contributions decreased with vital rate sensitivity (suggesting demographic buffering via stabilizing selection sensu Pfister 1998).

We have added text to P1 to help clarify our prediction here. [lines 112-122, 224-228] Although stabilizing selection occurs over time, we are comparing populations and we are using between-population comparisons. Saether and Bakke (2000) predicted smaller LTRE contributions (between populations) and other researchers have used “space-for-time” substitution with varying success (Strier 2016), so P1 uses the data available to see how well prospective elasticities predict observed (retrospective) vital rate contributions. We also mention in the Discussion how child survival actually varies a lot over time despite presumably strong stabilizing selection, but our main finding for P1 is that child survival differences have a smaller effect across populations than predicted by their high elasticities, whereas fertility differences have a larger effect. 

Main comment 4

I find that P1 (l81-82) is not well formulated. If I am not mistaken, it is a property of elasticity to be strictly declining with age in an age-structured model, infant and children survival elasticity always being constant and the largest. Metric have to be twisted and parameters very different that those of mammals to find alternative pattern (Baudish, 2005, PNAS). It is between species that relative magnitude of elasticities can be compared and I would strongly suggest to cite Heppell et al., 2000, Ecology for a comparison in mammals across the slow-fast continuum. Also why not refering to and using a classical Silvertown triangle to represented this (Silvertown, J., et al. 1993. Jouranl of Ecology 81:465–476)?

We now reference Heppel et al. 2000 when talking about age-patterns of elasticities differing within and between species [lines 131-133]. Whereas both the sensitivity and elasticity of survival decline with age due to declining expected future reproduction, elasticities are scaled by mean vital rates. This means that fertility elasticities are zero until AFB and then rise to a peak at some adult age, then decline due to mortality and and diminishing future fertility (Fig 1B).

We compare fitness contributions between populations and between species to see whether newborn survival (p0) makes the largest contribution because elasticity to recruitment is the largest elasticity. However, under strong buffering selection resulting in canalization, variation in newborn survival would be reduced, and thus fitness contributions from newborn survival should be small. Instead, we find large contributions of newborn survival, meaning that this vital rate is an important driver of population growth differences. This suggests that infant survival may not be buffered against variation, since Saether and Bakke (2000) predict smaller contributions from vital rates with high elasticities (Pfister 1998).

I am not sure what the authors want to test with prediction 2 which is the obvious fact that both increasing survival and fertility should increase population growth rate. Evidencing trade-off between fertility and survival?

Yes, exactly. Ordinarily you’d expect both higher survivorship and fertility to lead to higher growth rates – certainly this is the trivial case within a population, but does higher survivorship and fertility meaningfully predict higher growth rates across populations? They might not if survivorship benefits are among post-reproductive adults, or if there are trade-offs in vital rates. Such trade-offs have been documented in humans and non-human primates [24-28]. Our finding that population growth is decoupled from fertility in chimpanzees is consistent with a strong effect of mortality limiting potentially high fertility (if mortality were lower, high fertility would increase population growth more); that population growth is decoupled from survival in humans is consistent with long post-reproductive lifespans (contributing to high e0) during which direct fitness contributions are zero (surviving beyond ALB does not provide direct fitness).

As you suggest, we now include a ternary diagram (Fig 4), similar to those used by Silvertown 1993, as a more reader-friendly way of comparing elasticities for child survival, adult survival and fertility across populations. Fig 4A shows the ordination for elasticities (Ec, Ea, Ef), showing the strong selection on child survival and Fig 4B shows the ordination for vital rate effects (Cc, Ca, Cf), which are much more diverse and illustrate the difference between the potential for vital rate effects estimated by prospective elasticities and the observed vital rate effects, which scale elasticities by population-level differences.

Main comment 5

I would suggest the authors to discuss limitations of elasticities analyses in general and apply to humans in particular. (1) First elasticities are only one hand of the evolutionary GxE equation (Lande 1982;

Charlesworth 1990; Steppan et al. 2002). Evolution also need genetic variance and this could be

acknowledged. (2) The authors are comparing Leslie matrix, but any sub structuring (as individual

heterogeneity) or hidden trade-offs may change the results. (3) It the most important, it has been shown that intergenerational transfers between age-class or parental investment can strongly impacts elasticities on survival and fertility in humans (Lee 2003, PNAS, Pavard et al. 2007, Evolution, Pavard & Branger 2012 Theo Pop Biol). For instance, magnitude of elasticities on adult survival may be strongly underestimated when maternal or grand-maternal care is not implemented. Elasticities on fertilities by age can also exhibit very different patterns. Because such intergenerational transfers have been proposed as a very important drivers of the evolution of human life-history, the authors should at least discuss it. (4) As the authors wonderfully argued in a recent article, only periodic catastrophes in humans can explain the human forager paradox. It also means that all elasticity analysis in constant and infinite environment is somehow incomplete and elasticity should be considered into a stochastic model.

 We thank the reviewer for these insightful comments. 

(1) We now mention the need for genetic/phenotypic variability for natural selection to act on (and relate to canalization resulting small differences between populations, sensu Saether and Bakke 2000). [lines 516-517]

(2) Our asymptotic analyses look at the effects of changes in mean vital rates and do not address temporal variability in vital rates or the effects of individual variation. Although methods exist to decompose stochastic contributions (Davison et al. 2014), and we do reference temporal variation [lines 228-234] we do not have sufficient time-series date to conduct such analyses [lines 530-533].

(3) We mention the importance of intergenerational transfers driving indirect fitness contributions, and how intergenerational transfers can greatly alter the force of selection reflected in vital rate elasticities [lines 519-527]. However, estimating their fitness effects is not tractable with current methods, though we now cite Pavard et al. 2007 and Pavard & Branger 2012 for examples showing how transfers can impact elasticities in humans [lines 523-525]. We are excited to report that another paper in progress will present a new framework for estimating these indirect fitness contributions made via production or information transfers.

(4) Again, the existing subsistence population data limit our ability to conduct stochastic analyses but we acknowledge the importance of both demographic and environmental stochasticity that is missed in our analysis of averaged rates [lines 528-531]. We also miss individual heterogeneity that would be reflected in the underlying individual data but are obscured by our focus on average population statistics.

Minor comments & Détails

l418-420 – Isn’t there a contradiction is stating that juvenile survival is under canalization effect and stating later on that it varies more in time than adult survival?

Yes – if juvenile survival is canalized, then we shouldn’t expect it to vary much over time. In that case, our demonstration of substantial juvenile survival differences suggests that canalization is weak, or subject to environmental plasticity. We now clarify this point better in the paper to reduce any potential confusion [lines 224-235]. Juvenile survival is predicted to be canalized based on large elasticities but in fact it is highly variable (both over time as cited, and across populations as we show). This suggests limits on the ability of natural selection to buffer this rate (and we discuss why in terms of bet-hedging costs of reproduction).

Figure 1 – I am not sure how the SEM of elasticity is calculated. Is this trivial?

SEM = std(x)/sqrt(N), with std(x) taken across populations in a given set (e.g. hunter gatherers) and N being the number of populations in that set (e.g. 5 hunter-gatherers). 95% confidence intervals are represented by mean ± (2 SEM).

I am not sure what figure 2 really brings to the article since it is complicated and barely discussed.

Fig 2 summarizes the population-level net effects of survival vs. fertility. In the revised version, we explain its components better and discuss it more thoroughly in the paper [lines 336-343]. 

Problem in legend of figure 2 – Non-Forager are filled-circled as HG not filled square. Indicate that isolines are population growth rates. Remove the title.

Thanks for catching these. Typo corrected. Added reference to isoclines, and removed title. [lines 345-357]

L290 – I guess this is fig3.B instead of 2B,3?

It meant figure 2B and figure 3. Clarified in text. [lines 360, 364]

l 32 – I am not sure that reference [2] did anything in calculating the divergence time between humans and chimpanzees. Please check carefully this reference. I think it should instead be referred to l 33-34.

Thanks for your keen observation. We now more appropriately reference Hobolth et al. 2007 and Langergraber et al. 2012 there. [line 35]

l 35, “human fertility is similar to chimpanzees” and further. Please be more specific. Do you mean the shape of the age-specific fertilities? If yes, both the distribution and the TFR? Is the whole shape the distribution identical? The authors refer to [6] who focus mainly on reproductive senescence and show that if the timing of reproductive senescence is similar rate of reproductive senescence is not the same as well as how it correlates with decline in survival. I suggest to be more precise.

We added some remarks to better clarify similarities and differences in human and chimpanzee fertility [lines 37-40]. We cite [6] because of its supplement that includes ASFRs for multiple wild chimpanzee groups. In general, ASFR is similar during prime fertility ages but comparing them, on average, does reveal some key differences. Chimpanzees start reproducing at earlier ages and some continue to reproduce at later ages than humans (although most chimpanzees do not live until these later ages). In addition to fairly similar age profiles, which have been noted before, we find no significant difference in TFR, pointing out that completed chimpanzee fertility would be similar to that of human hunter-gatherers if they had human-like mortality schedules. 

L37-38 – “However, there is great variation among human and chimpanzee life histories”. Here again I suggest to be more specific. The authors quote [8]. Although a valid reference, it can be completed by more recent article (as the [2]). Furthermore, I am not a native speaker but is “difference” would be better than “variation”?

We add the reference [2], now [9] [line 42]. To clarify, we are looking here at the degree of variation in vital rates (shown in the SEM envelopes), which shows higher variation across populations among humans vs. chimpanzees. This variation is due to differences in many vital rates that drives differences in population growth rates. Whereas our LTRE contributions examine the effects of life history differences, here we are characterizing the degree of variation among our study populations.

L39 and many time after – Please change “within species” by between population. In ecology, within species study refers more to the study of variance between individuals than between population as it is investigate here.

We are not investigating individual-level variation but rather variation at the population level, within- vs. between- species. The variation across human populations is used to estimate the variation within humans, whereas species differences are investigated by comparing mean life histories of humans vs. chimpanzees. To avoid confusion, we clarify this distinction the first time it is introduced. [line 45-47]

L 40 - “We interpret population life histories in terms of the slow-fast life history continuum [9]” – Why? Also, human a complete outlier on this continuum so that I wonder if this is relevant.

Others have shown that primates are “slow” compared to other mammals, that chimpanzees are “slow” compared to other primates and that humans are “slow” compared to chimpanzees, so humans could well be seen as an outlier. As others have argued before us, even among humans there is a fast-slow continuum. At the same time that humans can be thought of as having “slow” life histories in comparative light, humans may also be considered outliers in the sense that they combine elements of slow life histories (longevity, delayed maturity) and elements of fast life histories (high fertility, short IBI). We cite other references for slow/fast LH in primates/chimpanzees/humans. [lines 42-45, 124-130, 519-523]

L41 – To my knowledge Stearns’ book (but I don’t have it at hand here to check) is about trade-offs in

general not about their importance for human life-history evolution.

True, we are extending the idea of life history tradeoffs to humans. Others (e.g. Gillespie et al. 2008; Lawson et al. 2012) show costs of reproduction in humans and so we cite that work here instead [lines 124-130, 512-515]

In [6] the authors use extensive data in chimpanzees. Yet, this represent only about 600-1000 individuals (the equivalent of a small human village) spread in small groups over nearly a continent. How this could affect the authors’ results?

The small sample sizes may not accurately represent the diversity of chimpanzee life histories (or the diversity of individual life courses among chimpanzees), but they are the best data available. The ASFRs are fairly similar across groups. Thus, in [6] the authors provide ASFRs in two ways: averages across populations, and combined as if all from the same population (i.e. equivalent to weighted by sample size). The differences in ASFRs across these two conditions are minimal. 

L55 – I am not found of the concept of population fitness underlying in this sentence.

Does this mean the reviewer is skeptical about the utility of elasticities for estimating the force of selection? This has a long tradition, but there is also a debate of the usefulness of sensitivities vs. elasticities

Elasticities and sensitivities are both useful in that they tell us about the potential for fitness effects if vital rates change (a measure of “bang-for-your-buck” in terms of fitness changes when vital rates differ). Elasticities have the additional utility of being proportional (a % increase in population growth due to a % increase in a vital rate); by scaling differences by their mean values elasticities allow us to compare the relative effects of changes in fertility vs. survival on the same y-axis. LTRE contributions, on the other hand, scale fitness effects to the differences between populations, so small differences in a high-elasticity rate can make larger fitness contributions than large differences in a low-elasticity rate. LTRE contributions, therefore, show us what actually explains differences in population growth rates based on the observed vital rate differences and their associated elasticities. Comparing realized (retrospective) LTRE contributions to potential suggested by (prospective) elasticities may tell us about potential constraints on selection when vital rate variation is limited (due to inherently low genetic variation or stabilizing selection being stronger than directional selection).

L71 – Indicates the pages in [15]. Note that you could have also quoted Hal Caswell, 1989, Analysis of life table response experiments I. Decomposition of effects on population growth rate, Ecological Modelling, Volume 46, Issues 3–4.

We now cite Caswell 1989 [line 121]

L153 – this should be sij instead of sj isn’t it?

Typo fixed [line 186]

L179 – I am not sure to understand why the fact that Cij and Eij sum to unity allow to calculate the ratio.

Because Cij and Eij each sum to unity, they represent the proportion of all observed effects and the proportion of potential effects, respectively. Therefore, we can sum the proportion of effects due to child survival and the proportion of elasticities to child survival and see if a larger or smaller proportion of retrospective effects was observed, relative to the potential suggested by prospective elasticities. This contrast is now illustrated by a new Fig 4.

We acknowledge that ratios of ratios are sometimes confusing or inappropriate, but here the ratio tells us whether proportional LTRE contributions due to X are greater than the proportional elasticity effects due to X. Because we already examine the net LTRE contributions (Fig. 3A), we look at effect magnitudes, which do not obscure opposing contributions (C < 0 vs C > 0), and we scale each effect magnitudes relative to the sum of all effect magnitudes because this makes them directly comparable to elasticities (which are likewise scaled to sum to unity across all the potential effects). [lines 205-210].

Figure 3A – I find the figure very complicated to figure out. Are they the mean summed C values between populations? Then why and how is separated positive and negative C values? Or “composite” refers to the mean trajectories for HG, F, WC. But then, again, how does it lead to both (+) and (-) for a same trait (i.e., Infant survival). I am very sorry if I miss this information.

Fig 3A shows net results for contributions (summing to the difference between a population growth rate and that of the mean hunter-gatherer reference), vs. 3B showing the relative effect magnitudes (summing to unity to be comparable with elasticities – see the new Fig 4). These effect magnitudes are what we use to test for differences in means and to calculate the Z ratios (for P1).

In Fig 3A, the positive values are summed above the origin and the negative values summed below, illustrating opposing contributions that, on net, yield the difference in population growth rate (inset white bars). For instance, Infant survival is from age 0 to age 2 so there are two age-contributions (a21 for survival 0 to 1, a32 for survival 1 to 2) that may be opposite in sign. We make a note in the figure caption explaining why contributions for a given life cycle component (e.g. infant survival) can have opposing contributions above and below the origin (C = 0). [lines 370-373]

The “composite” populations in Fig. 3A are the synthetic populations with vital rates equal to the average over a given group (e.g. the LTRE reference life history with vital rates averaged across hunter-gatherers, HG). Fig. 3B legend now clarifies that effect magnitudes averaged across groups of individual populations (e.g. the 5 hunter-gatherer populations), rather than results for the mean life histories (e.g. the mean hunter-gatherer reference) [lines 377-380].

Fig 3A shows vital rate contributions that are summed across life history components (infant, child and adult survival; early, prime and late fertility), but the positive and negative components are summed and plotted separately (positive contributions above zero and negative contributions upside-down below zero). If individual vital rates within a single life history component may have different signs they will appear both above and below the zero line (e.g. infant survival is from age zero to age 2 and includes both newborn survival age 0 to 1 and survival age 1 to 2, so it may have both positive and negative contributions). In Fig 3B these opposing contributions are weighted equally in the combined magnitude so they sum to unity (100% of all effect magnitudes).

---

## [Editor Report · Decision Letter 2]

28 Jan 2021

Human uniqueness? Life history diversity among small-scale societies and chimpanzees

PONE-D-20-26560R2

Dear Dr. Davison,

We’re pleased to inform you that your manuscript has been judged scientifically suitable for publication and will be formally accepted for publication once it meets all outstanding technical requirements.

Kind regards,

Masami Fujiwara, PhD

Academic Editor

PLOS ONE
---

## [Editor Report · Acceptance letter]

8 Feb 2021

PONE-D-20-26560R2 

Human uniqueness? Life history diversity among small-scale societies and chimpanzees 

Dear Dr. Davison:

I'm pleased to inform you that your manuscript has been deemed suitable for publication in PLOS ONE. Congratulations! Your manuscript is now with our production department. 

Kind regards, 

on behalf of

Dr. Masami Fujiwara 

Academic Editor

PLOS ONE